# Stochastic processes constrain the within and between host evolution of influenza virus

John T McCrone[1], Robert J Woods[2], Emily T Martin[3], Ryan E Malosh[3], Arnold S Monto[3], Adam S Lauring[1,2]*

[1]Department of Microbiology and Immunology, University of Michigan, Ann Arbor, United States; [2]Division of Infectious Diseases, Department of Internal Medicine, University of Michigan, Ann Arbor, United States; [3]Department of Epidemiology, University of Michigan, Ann Arbor, United States

**Abstract** The evolutionary dynamics of influenza virus ultimately derive from processes that take place within and between infected individuals. Here we define influenza virus dynamics in human hosts through sequencing of 249 specimens from 200 individuals collected over 6290 person-seasons of observation. Because these viruses were collected from individuals in a prospective community-based cohort, they are broadly representative of natural infections with seasonal viruses. Consistent with a neutral model of evolution, sequence data from 49 serially sampled individuals illustrated the dynamic turnover of synonymous and nonsynonymous single nucleotide variants and provided little evidence for positive selection of antigenic variants. We also identified 43 genetically-validated transmission pairs in this cohort. Maximum likelihood optimization of multiple transmission models estimated an effective transmission bottleneck of 1–2 genomes. Our data suggest that positive selection is inefficient at the level of the individual host and that stochastic processes dominate the host-level evolution of influenza viruses.

DOI: https://doi.org/10.7554/eLife.35962.001

*For correspondence:
alauring@med.umich.edu

Competing interests: The authors declare that no competing interests exist.

## Introduction

The rapid evolution of influenza viruses has led to reduced vaccine efficacy and the continuing emergence of novel strains. Broadly speaking, evolution is the product of deterministic processes, such as selection, and stochastic processes, such as genetic drift and migration (*Kouyos et al., 2006*). The global evolution of influenza A virus (IAV) is dominated by the positive selection of novel antigenic variants that subsequently seed annual epidemics in the Northern and Southern hemisphere (*Rambaut et al., 2008*). These global patterns are contrasted by results from whole genome sequencing of viruses on more local scales, which suggest the importance of stochastic processes such as strain migration and within-clade reassortment (*Nelson et al., 2006*). While it is clear that influenza's evolutionary dynamics differ across scales (*Nelson and Holmes, 2007*; *Holmes, 2009*), the relative roles of deterministic and stochastic forces in viral evolution within and between acutely infected individuals remain unclear.

It is now feasible to efficiently sequence patient-derived isolates at sufficient depth of coverage to define the genetic diversity and dynamics of virus evolution within individual hosts (*Kao et al., 2014*). Studies of IAV populations in animal and human systems suggest that most intrahost single nucleotide variants (iSNV) are rare and that intrahost populations are subject to strong purifying selection (*Rogers et al., 2015*; *Murcia et al., 2010*; *Iqbal et al., 2009*; *Poon et al., 2016*; *Dinis et al., 2016*; *Debbink et al., 2017*). Positive selection is common in cell culture (*Doud et al., 2017*; *Archetti and Horsfall, 1950*; *Foll et al., 2014*), and has been observed in experimental

infections in swine (*Illingworth et al., 2014*). However, it has only been well documented within human hosts in the extreme cases of drug resistance (*Gubareva et al., 2001*; *Ghedin et al., 2011*; *Rogers et al., 2015*), and long-term infection of immunocompromised individuals (*Xue et al., 2017*). Indeed, we and others have been unable to identify evidence for positive selection in naturally-occurring, acute human infections (*Debbink et al., 2017*; *Dinis et al., 2016*), and its relevance to within-host processes is unclear.

Despite limited evidence for positive selection, novel mutations do arise within hosts and some will clearly be positively selected. Their potential for subsequent spread through host populations is heavily dependent on the size of the transmission bottleneck (*Alizon et al., 2011*; *Zwart and Elena, 2015*). If the transmission bottleneck is sufficiently wide, low frequency variants can plausibly be transmitted and spread efficiently through host populations (*Geoghegan et al., 2016*). While experimental infections of ferrets suggest a very narrow transmission bottleneck (*Varble et al., 2014*; *Wilker et al., 2013*), studies of equine influenza support a bottleneck wide enough to allow transmission of rare iSNV (*Hughes et al., 2012*; *Murcia et al., 2010*). The only available genetic study of influenza virus transmission in humans estimated a remarkably large transmission bottleneck, allowing for transmission of 100–200 genomes (*Poon et al., 2016*; *Sobel Leonard et al., 2017*).

Here, we use next generation sequencing of within-host influenza virus populations to define the evolutionary dynamics of IAV within and between human hosts. We apply a benchmarked analysis pipeline to identify iSNV and to characterize the genetic diversity of H3N2 and H1N1 populations collected over five post-pandemic seasons from individuals enrolled in a prospective household study of influenza. We find that intrahost populations are dynamic and constrained by genetic drift and purifying selection. In our study, positive selection rarely amplifies a beneficial de novo variant to a frequency greater than 2%. Contrary to what has been previously reported for human influenza transmission (*Poon et al., 2016*), but consistent with what has been observed in many other viruses with distinct modes of transmission (*Zwart and Elena, 2015*; *McCrone and Lauring, 2018*), we identify a very tight effective transmission bottleneck that limits the transmission of low-frequency variants.

## Results

We used next generation sequencing to characterize influenza virus populations collected from individuals enrolled in the Household Influenza Vaccine Effectiveness (HIVE) study (*Monto et al., 2014*; *Ohmit et al., 2013*; *Ohmit et al., 2015*; *Ohmit et al., 2016*; *Petrie et al., 2013*), a community-based cohort that enrolls 213–340 households of 3 or more individuals in Southeastern Michigan each year (*Table 1*). These households are followed prospectively from October to April, with symptom-triggered collection of nasal and throat swab specimens for identification of respiratory viruses by RT-PCR (see Materials and methods). In contrast to case-ascertained studies, which identify households based on an index case who seeks medical care, the HIVE study identifies symptomatic individuals regardless of illness severity. In the first four seasons of the study (2010–2011 through 2013–2014), respiratory specimens were collected 0–7 days after illness onset. Beginning in the 2014–2015 season, each individual provided two samples, a self-collected specimen at the time of symptom onset and a clinic-collected specimen obtained 0–7 days later. Each year, 59–69% of individuals had self-reported or confirmed receipt of that season's vaccine prior to local circulation of influenza virus.

Over five seasons and nearly 6290 person-seasons of observation, we identified 77 cases of influenza A/H1N1pdm09 infection and 313 cases of influenza A/H3N2 infection (*Table 1*). Approximately half of the cases (n = 166) were identified in the 2014–2015 season, in which there was an antigenic mismatch between the vaccine and circulating strains (*Flannery et al., 2016*). All other seasons were antigenically matched. Individuals within a household were considered an epidemiologically linked transmission pair if they were both positive for the same subtype of influenza virus within 7 days of each other. Several households had 3 or four symptomatic cases within this one-week window, suggestive of longer chains of transmission (*Table 1*).

### Within-host populations have low genetic diversity

We processed all specimens for viral load quantification and next generation sequencing. Viral load measurements (genome copies per μl) were used for quality control in variant calling, which we have

**Table 1.** Influenza viruses over five seasons in a household cohort

| | 2010–2011 | 2011–2012 | 2012–2013 | 2013–2014 | 2014–2015 |
|---|---|---|---|---|---|
| Households | 328 | 213 | 321 | 232 | 340 |
| Participants | 1441 | 943 | 1426 | 1049 | 1431 |
| Vaccinated, n (%)* | 934 (65) | 554 (59) | 942 (66) | 722 (69) | 992 (69) |
| IAV Positive Individuals[†] | 86 | 23 | 69 | 48 | 166 |
| H1N1 | 26 | 1 | 3 | 47 | 0 |
| H3N2 | 58 | 22 | 66 | 1 | 166 |
| IAV Positive Households[‡] | | | | | |
| Two individuals | 13 | 2 | 9 | 7 | 23 |
| Three individuals | 5 | 2 | 3 | 3 | 11 |
| Four individuals | - | - | 1 | 2 | 4 |
| High Quality NGS Pairs[§] | 4 | 1 | 2 | 6 | 39 |

*Self reported or confirmed receipt of vaccine prior to the specified season.

[†]RT-PCR confirmed infection.

[‡]Households in which two individuals were positive within 7 days of each other. In cases of trios and quartets, the putative chains could have no pair with onset > 7 days apart.

[§]Samples with > $10^3$ genome copies per μl of transport medium, adequate amplification of all eight genomic segments, and average sequencing coverage > $10^3$ per nucleotide.

DOI: https://doi.org/10.7554/eLife.35962.002

shown is highly sensitive to input titer (*McCrone and Lauring, 2016*) (*Figure 1A*). Accordingly, we report data on 249 high quality specimens from 200 individuals, which had a viral load of >$10^3$ copies per microliter of transport media, adequate RT-PCR amplification of all eight genomic segments, and an average read coverage of >$10^3$ across the genome (*Table 1*, *Figure 1—figure supplement 1*).

We identified intrahost single nucleotide variants (iSNV) using our empirically validated analysis pipeline (*McCrone and Lauring, 2016*). Our approach relies heavily on the variant caller DeepSNV, which uses a clonal plasmid control to distinguish between true iSNV and errors introduced during sample preparation and/or sequencing (*Gerstung et al., 2012*). Given the diversity of influenza viruses that circulate locally each season, there were a number of instances in which our patient-derived samples had mutations that were essentially fixed (>0.95 frequency) relative to the clonal control. DeepSNV is unable to estimate an error rate for the control or reference base at these positions. We therefore performed an additional benchmarking experiment to identify a threshold for majority iSNV at which we could correctly infer whether or not the corresponding minor allele was also present (see Materials and methods). We found that we could correctly identify a minor allele at a frequency of ≥2% at such sites. We therefore report data on iSNV present at frequencies between 2% and 98%. As expected, this threshold improved the specificity of our iSNV identification and decreased our sensitivity to detect variants below 5% compared to our initial validation experiment (*McCrone and Lauring, 2016*), which did not employ a frequency threshold (*Supplementary file 1*).

Consistent with our previous studies of natural infections and those of others, we found that the within-host diversity of seasonal influenza A virus (IAV) populations is low (*Dinis et al., 2016*; *Debbink et al., 2017*; *Sobel Leonard et al., 2016*; *McCrone and Lauring, 2016*). Two hundred forty-three out of the 249 samples had fewer than 10 minority iSNV (median 2, IQR 1–3). There were six samples with greater than 10 minority iSNV. In 3 of these cases, the frequency distribution of minority iSNVs was bimodal, suggesting that the iSNV were linked and that the samples represented mixed infections. Consistent with this hypothesis, putative genomic haplotypes based on these minority iSNV clustered with distinct isolates on phylogenetic trees (*Figure 1—figure supplements 2* and *3*), and these samples were removed from subsequent analysis. While viral shedding was well correlated with days post symptom onset (*Figure 1A*), the number of minority iSNV identified was not affected by the day of infection, viral load, subtype, or vaccination status (*Figure 1B and C*, and

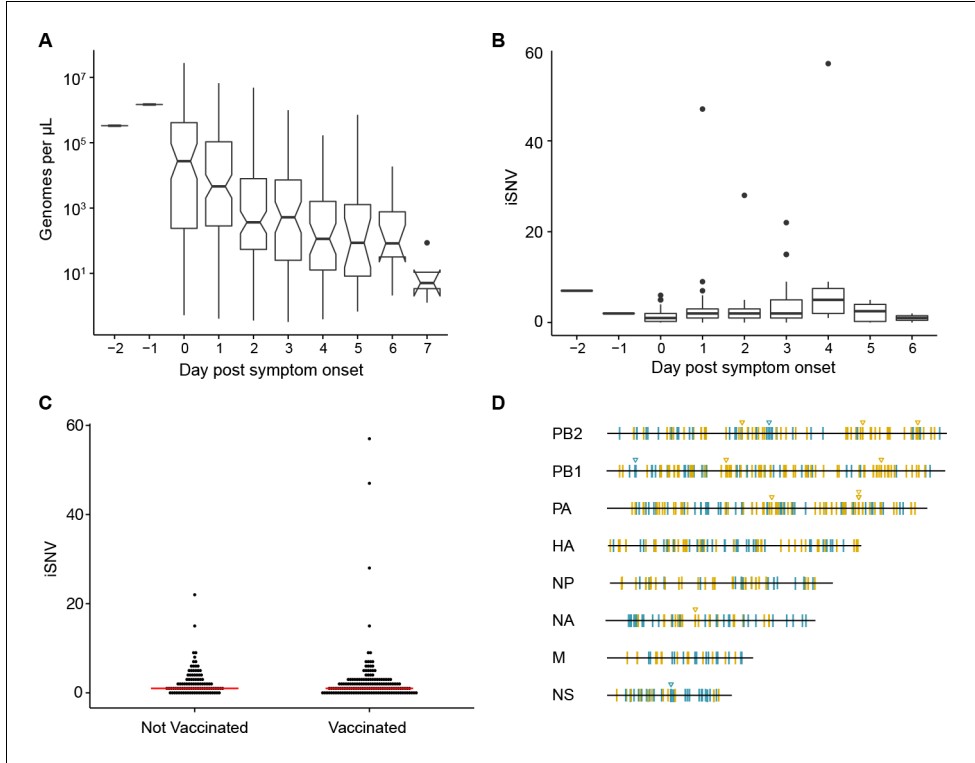

**Figure 1.** Within-host diversity of IAV populations. (**A**) Boxplots (median, 25th and 75th percentiles, whiskers extend to most extreme point within median ±1.5 x IQR) of the number of viral genomes per microliter transport media stratified by day post symptom onset. Notches represent the approximate 95% confidence interval of the median. (**B**) Boxplots (median, 25th and 75th percentiles, whiskers extend to most extreme point within median ±1.5 x IQR) of the number of iSNV in 249 high quality samples stratified by day post symptom onset. (**C**) The number of iSNV in each isolate stratified by vaccination status. The red lines indicate the median. (**D**) Location of all identified iSNV in the influenza A genome. Mutations are colored nonsynonymous (blue) and synonymous (gold) relative to that sample's consensus sequence. Mutations are considered nonsynonymous if they are nonsynonymous in any known influenza open reading frame. Triangles signify mutations that were found in more than one individual in a given season.

DOI: https://doi.org/10.7554/eLife.35962.003

The following source data and figure supplements are available for figure 1:

**Source data 1.** Titers and day of sampling for all samples processed from the cohort.
DOI: https://doi.org/10.7554/eLife.35962.008
**Source data 2.** The number of iSNV and day of sampling for samples that qualified for iSNV identification.
DOI: https://doi.org/10.7554/eLife.35962.009
**Source data 3.** The number of iSNV and vaccination status for samples that qualified for iSNV identification.
DOI: https://doi.org/10.7554/eLife.35962.010
**Source data 4.** Location and frequency of iSNV identified in each individual.
DOI: https://doi.org/10.7554/eLife.35962.011
**Figure supplement 1.** Sequence coverage for all samples.
DOI: https://doi.org/10.7554/eLife.35962.004
**Figure supplement 2.** Approximate maximum likelihood trees of the concatenated coding sequences for high quality H1N1 samples.
DOI: https://doi.org/10.7554/eLife.35962.005
**Figure supplement 3.** Approximate maximum likelihood trees of the concatenated coding sequences for high quality H3N2 samples.
DOI: https://doi.org/10.7554/eLife.35962.006
**Figure supplement 4.** The effect of titer on the number of iSNV identified.
DOI: https://doi.org/10.7554/eLife.35962.007

*Figure 1—figure supplement 4*). Single nucleotide variants were distributed evenly across the genome (*Figure 1D*).

Minority variants were rarely shared among multiple individuals. Ninety-eight percent of minority iSNV were only found once, 2.3% were found in two individuals, and no minority iSNV were found in three or more individuals. The low level of shared diversity suggests that within-host populations explore distinct regions of sequence space with little evidence for parallel evolution. Of the 12 minority iSNV that were found in multiple individuals (triangles in *Figure 1D*), three were nonsynonymous and none were present in antigenic epitopes. The vast majority of minority variants were rare (frequency 0.02–0.07) (*Figure 2A*). The ratio of nonsynonymous to synonymous variants was 0.75, and given the excess of nonsynonymous sites across the genome and within the HA gene, these data suggest significant purifying selection within hosts.

## iSNV within and outside antigenic epitopes have similar frequencies

Although the full range of the H3 antigenic sites have not been functionally defined, it is estimated that 131 of the 329 amino acids in HA1 lie in or near these sites (*Lee and Chen, 2004*). Although we identified 16 minority nonsynonymous iSNV in these regions (*Supplementary file 2*), mutations in antigenic epitopes were not present at significantly higher within-host frequencies than other nonsynonymous mutations in HA (*Figure 2B*). Five putative antigenic variants were in positions that have been experimentally shown to differ among antigenically drifted viruses (*Smith et al., 2004*; *Wiley et al., 1981*), and two (193S and 189N) lie in the 'antigenic ridge' that is a major contributor to antigenic drift (*Koel et al., 2013*). Five of the iSNV in antigenic epitopes have been detected as minority variants above 5% at the global level since the time of isolation (62G, 128A, 193S, 262N, and 307R) with one (62G) seemingly increasing in global frequency (*Figure 2C*). An additional within host variant (307R) has dominated the population for the past 12 years (*Neher and Bedford, 2015*). We identified one putative H1N1 antigenic variant (208K in antigenic site $C_a$) (*Caton et al., 1982*; *Xu et al., 2010*). In total, putative antigenic variants account for 1.0–2.8% of minority iSNV identified and were found in 3.0–7.1% of infections. None of these iSNV were shared among multiple individuals.

## Within host populations are dynamic

We next examined the changes in within-host virus diversity over time in 49 individuals who provided paired longitudinal samples during the 2014/2015 season (*Figure 2D*). We found that there was very little change in iSNV frequency in populations sampled twice on the same day ($R^2$ = 0.982, *Figure 2E* and *Figure 2—figure supplement 1A*), and the concordance of same day samples suggests that our sampling procedure is reproducible. An analysis of sequence data from replicate libraries indicates that our estimates of iSNV frequency are precise (*Figure 2—figure supplement 1B*). In samples separated by at least a day, only 57% iSNV found in the first sample persisted in the second, above the 2% limit of detection. Additionally, the majority of iSNV (68%) found in the second sample were either new or previously present below the 2% limit of detection. Taken together, these data suggest that the population present in the upper respiratory tract is highly dynamic while maintaining a stable consensus. Of note, 6 iSNV in our longitudinal data set lay within antigenic epitopes (arrows in *Figure 2E*). Their dynamics are similar to those of other nonsynonymous iSNV and synonymous iSNV, suggesting that most mutations change in frequency due to stochastic as opposed to selective processes. Together with our prior work (*Debbink et al., 2017*), these data suggest that the positive selection of novel variants within hosts is inefficient and rarely amplifies a newly generated variant to a frequency greater than 2%.

## Identification of forty-three transmission pairs

Our within-host data suggest that newly arising iSNV with positive fitness effects are likely to be present at low frequencies (<2%) during an acute infection. The maintenance of these mutations in host populations is therefore highly dependent on the transmission bottleneck. We analyzed virus populations from 85 households with concurrent infections to quantify the level of shared viral diversity and to estimate the size of the IAV transmission bottleneck (*Table 1*). Because epidemiological linkage does not guarantee that concurrent cases constitute a transmission pair (*Petrie et al., 2017*), we used stringent criteria to eliminate individuals in a household with co-incident community

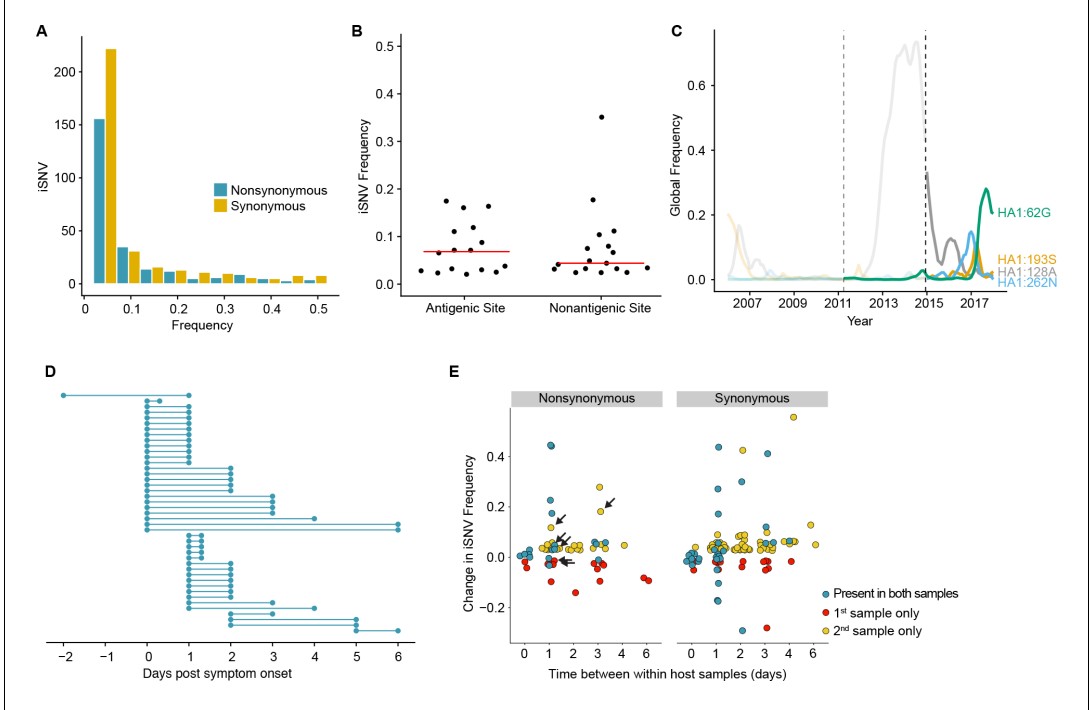

**Figure 2.** Within-host dynamics of IAV. (A) Histogram of within-host iSNV frequency in 249 high quality samples. Bin width is 0.05 beginning at 0.02. As in *Figure 1*, mutations were classified as nonsynonymous (blue) if they were nonsynonymous in any known influenza reading frame. Synonymous mutations are gold. (B) The within-host frequency of nonsynonymous mutations in HA stratified by whether or not they are in known antigenic sites (p=0.46 Wilcoxon rank sum). (C) The global frequency of putative antigenic minority iSNV identified in our cohort that have circulated at frequencies above 5% globally since their time of collection. Each variant is labeled according the H3 numbering scheme. The dashed line indicates when samples were collected. Frequency traces are faded prior to the collection date. (D) Timing of sample collection for 43 paired longitudinal samples relative to day of symptom onset. Of the 49 total, 43 pairs had minority iSNV present in either sample. (E) The change in frequency over time for minority iSNV identified for the paired samples in (A). Nonsynonymous and synonymous iSNV are plotted separately. Mutations are colored according to whether they were detected in both isolates (blue), detected only the first isolate (red), or detected only in the second isolate (yellow). The threshold of detection was 2%. The arrows indicate mutations in known antigenic sites.

DOI: https://doi.org/10.7554/eLife.35962.012

The following source data and figure supplement are available for figure 2:

**Source data 1.** The frequency and class (nonsynonymous/synonymous) of identified iSNV.
DOI: https://doi.org/10.7554/eLife.35962.014
**Source data 2.** Meta data for nonsynonymous iSNV found in HA.
DOI: https://doi.org/10.7554/eLife.35962.015
**Source data 3.** Frequency and meta data for antigenic iSNV that were also identified at the global level.
DOI: https://doi.org/10.7554/eLife.35962.016
**Source data 4.** Sampling day for within-host sample pairs.
DOI: https://doi.org/10.7554/eLife.35962.017
**Source data 5.** Frequencies of mutations identified in longitudinal sample pairs.
DOI: https://doi.org/10.7554/eLife.35962.018
**Figure supplement 1.** (A) Reproducibility of iSNV identification for paired samples acquired on the same day.
DOI: https://doi.org/10.7554/eLife.35962.013

acquisition of distinct viruses. We considered all individuals in a household with symptom onset within a 7 day window to be epidemiologically linked. The donor in each putative pair was defined as the individual with the earlier onset of symptoms. We discarded a transmission event if there were multiple possible donors with the same day of symptom onset. Donor and recipients were not allowed to have symptom onset on the same day, unless the individuals were both index cases for the household. In these six instances, we analyzed the data for both possible donor-recipient directionalities. Based on these criteria, our cohort had 124 putative household transmission events over

five seasons (*Table 1*). Of these, 52 pairs had samples of sufficient quality for reliable identification of iSNV from both individuals.

We next used sequence data to determine which of these 52 epidemiologically linked pairs represented true household transmission events as opposed to coincident community-acquired infections. We measured the genetic distance between influenza populations from each household pair by L1-norm and compared these distances to those of randomly assigned community pairs within each season (*Figure 3A*, see also trees in *Figure 1—figure supplements 2* and *3*). While the L1-norm of a pair captures differences between the populations at all levels, in our cohort, it was largely driven by differences at the consensus level. We only considered individuals to be a true transmission pair if they had a genetic distance below the 5th percentile of the community distribution of randomly assigned pairs (*Figure 3A*). Forty-seven household transmission events met this criterion (*Figure 3B*). Among these 47 sequence-validated transmission pairs, three had no iSNV in the donor and one additional donor appeared to have a mixed infection. These four transmission events were removed from our bottleneck analysis, as donors without iSNV are uninformative and mixed infections violate model assumptions of site independence (see Materials and methods). We estimated the transmission bottleneck in the remaining 43 high-quality pairs (37 H3N2, 6 H1N1, *Figure 3B*).

A transmission bottleneck restricts the amount of genetic diversity that is shared by both members of a pair. We found that few minority iSNV where polymorphic in both the donor and recipient populations (*Figure 3C*). Minority iSNV in the donor were either absent or fixed in the recipient (top and bottom of plot). The lack of shared polymorphic sites (which would lie in the middle of the plot in *Figure 3C*) suggests a stringent effective bottleneck in which only one allele is passed from donor to recipient.

## Estimation of the transmission bottleneck

We applied a simple presence-absence model to quantify the effective transmission bottleneck in our cohort. The presence-absence model considers only whether or not a donor allele is present or absent in the recipient sample. Under this model, transmission is a neutral, random sampling process, and the probability of transmission is simply the probability that the iSNV will be included at least once in the sample given its frequency in the donor and the sample size, or bottleneck. We estimated a distinct bottleneck for each transmission pair and assumed these bottlenecks followed a zero-truncated Poisson distribution. Maximum likelihood optimization determined that a mean bottleneck of 1.68 (lambda = 1.15; 0.49–2.14, 95% CI) best described the data. This distribution indicates that the majority of bottlenecks are 1 and that 95% of bottlenecks are expected to be ≤3. One outlier was a single transmission pair with an estimated bottleneck of >200; this pair had two minority iSNV present at very low frequencies in both donor and recipient. There were no apparent differences between H3N2 and H1N1 pairs. The model fit was evaluated by comparing the probability of transmission predicted by the model with that estimated from the data using a sliding window (*Figure 3D*). The predicted transmission probabilities capture the underlying trend in the data, which suggests that transmission dynamics can be appropriately modeled using a selectively neutral, stringent bottleneck.

Because the presence-absence model can underestimate the transmission bottleneck in some circumstances (*Sobel Leonard et al., 2017*), we also applied a beta binomial model, which Leonard *et al.* have used to account for the stochastic dynamics of transmitted variants. This model allows for a limited amount of time-independent genetic drift within the recipient (*Sobel Leonard et al., 2017*), and we modified it to also account for our benchmarked sensitivity for rare variants (*Supplementary file 1*, Current Pipeline). For all donor-derived iSNV that were absent in the recipient, we estimated the likelihood that these variants were transmitted but either drifted below our level of detection or drifted below 10% and were missed by our variant identification. Despite the relaxed assumptions provided by this modified beta binomial model, maximum likelihood estimation only marginally increased the average bottleneck size (mean 1.75: lambda 1.25; 0.55–2.28, 95% CI) relative to the simpler presence-absence model. We also fit a long-tailed, discrete distribution based on the log-normal. As expected, this analysis resulted in a slightly wider distribution with a mode of 1, a 95[th] percentile of 11 and a 97.5th percentile of 21. Although the beta-binomial again reproduced the trend in the data (*Figure 3E*), the fit was not better than that of the presence-absence model (AIC 80.2 for beta-binomial compared to 74.5 for the presence-absence model), suggesting the stochastic loss of iSNV in the recipient is not a major contributing factor to the transmission

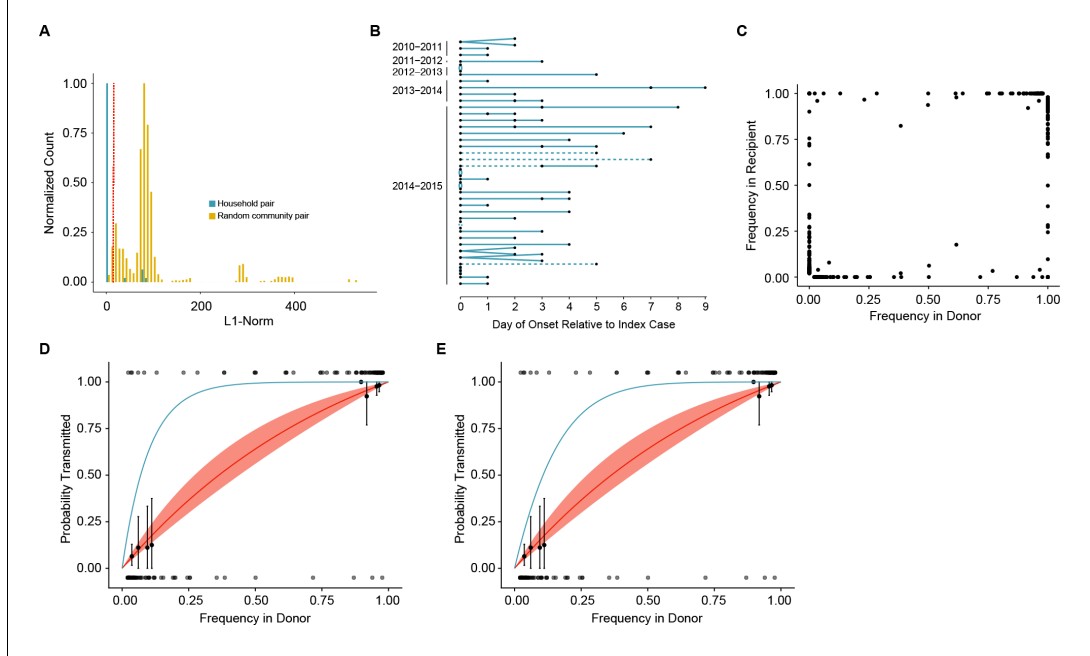

**Figure 3.** Between host dynamics of IAV. (**A**) The distribution of pairwise L1-norm distances for household (blue) and randomly-assigned community (gold) pairs. The bar heights are normalized to the height of the highest bar for each given subset (47 for household, 1592 for community). The red line represents the 5th percentile of the community distribution. (**B**) Timing of symptom onset for 52 epidemiologically linked transmission pairs. Days of symptom onset for both donor and recipient individuals are indicated by black dots. Dashed lines represent pairs that were removed due to abnormally high genetic distance between isolates, see (**A**). (**C**) The frequency of donor iSNV in both donor and recipient samples. Frequencies below 2% and above 98% were set to 0% and 100% respectively. (**D**) The presence-absence model fit compared with the observed data. The x-axis represents the frequency of donor iSNV with transmitted iSNV plotted along the top and nontransmitted iSNV plotted along the bottom. The line represents the predicted probability of transmission given by the presence-absence model with a mean bottleneck of 1.68. The shaded regions represent the 95% confidence interval. Black points on the plot represent the probability of transmission estimated as the proportion of iSNV transmitted within a sliding window of width 5% and a step of 1%. The error bars represent the 95% confidence interval and were derived from a binomial distribution as in (*Sobel Leonard et al., 2017*). Only those windows with more than 5 iSNV are plotted. Blue curve shows the probability of transmission at a given frequency given a bottleneck size of 10 in the presence-absence model. (**E**) The beta-binomial model fit. Similar to (**D**), except the predicted outcomes are the based on a beta-binomial model using a mean bottleneck of 1.75. Blue curve shows the probability of transmission at a given frequency given a bottleneck size of 10 in the beta-binomial model.

DOI: https://doi.org/10.7554/eLife.35962.019

The following source data and figure supplement are available for figure 3:

**Source data 1.** Genetic distance of household and community sample pairs.
DOI: https://doi.org/10.7554/eLife.35962.021

**Source data 2.** Day of onset and meta data for transmission pairs.
DOI: https://doi.org/10.7554/eLife.35962.022

**Source data 3.** Frequencies of iSNV identified in transmission pairs.
DOI: https://doi.org/10.7554/eLife.35962.023

**Source data 4.** The model prediction of the probability of transmission given donor frequency for the presence-absence model.
DOI: https://doi.org/10.7554/eLife.35962.024

**Source data 5.** The frequency of donor iSNV used in fitting transmission models.
DOI: https://doi.org/10.7554/eLife.35962.025

**Source data 6.** The model prediction of the probability of transmission given donor frequency for the beta-binomial model.
DOI: https://doi.org/10.7554/eLife.35962.026

**Source data 7.** Bottleneck estimates for all isolate pairings in cases where multiple donor or recipient isolates are available.
DOI: https://doi.org/10.7554/eLife.35962.027

**Figure supplement 1.** Estimate of effective bottleneck size with relaxed variant calling criteria.
DOI: https://doi.org/10.7554/eLife.35962.020

dynamics observed here. As in the presence-absence model, there was no apparent stratification in bottleneck size based on vaccination status, donor or recipient age, or influenza A subtype (*Supplementary file 3*).

Because our bottleneck estimates were much lower than what has previously been reported for human influenza (*Poon et al., 2016*), we investigated the impact that our simplifying assumptions could have on our results. In particular, the presence-absence model assumes perfect detection of variants in donor and recipient, and it can therefore underestimate the size of a bottleneck in the setting of donor-derived variants that are transmitted but not detected in the recipient. These 'false negative' variants can occur when the frequency of an iSNV drifts below the level of detection (e.g. 2% frequency) or when the sensitivity of sequencing is less than perfect for variants at that threshold (e.g. 15% sensitivity for variants at a frequency 2–5%). To determine the impact of sequencing sensitivity and specificity on our bottleneck estimates, we re-called variants using our original pipeline without the 2% frequency cut-off. As shown in *Supplementary file 1*, this increases the sensitivity of iSNV detection in the 1–5% frequency range, and also the number of false positive variant calls (*McCrone and Lauring, 2016*). This analysis only slightly increased average transmission bottleneck to 2.11 (lambda = 1.75; 0.87–2.95, 95% CI), and indicates that our results are not biased by the added stringency used in the initial analysis. Interestingly, the inclusion of these low-frequency iSNV reduced model fit and led to an overestimation of the probability of transmitting rare iSNV (*Figure 3—figure supplement 1*). These low-frequency bins are likely dominated by iSNV that were not present at the time of transmission either because they are sequencing artifacts or underwent stochastic extinction in the donor prior to transmission.

## Discussion

We characterized influenza virus dynamics in human hosts through sequencing of 249 specimens from 200 individuals collected over 6290 person-seasons of observation. In our study, we find that acute influenza infections are characterized by low diversity, limited positive selection, and tight transmission bottlenecks. Because we used viruses collected over five influenza seasons from individuals enrolled in a prospective household cohort, these dynamics are likely to be broadly representative of many seasonal influenza infections in temperate regions. Our results are further strengthened by the use of a validated sequence analysis pipeline and models that are robust to the underlying assumptions. The observed within-host dynamics and the tight effective transmission bottleneck suggest that stochastic processes, such as genetic drift, dominate influenza virus evolution at the level of individual hosts. This stands in contrast to the prominent role of positive selection in the global evolution of seasonal influenza.

Contrary to previous studies, which have found signatures of adaptive evolution in infected hosts (*Gubareva et al., 2001*; *Rogers et al., 2015*; *Ghedin et al., 2011*; *Sobel Leonard et al., 2016*), we have found only limited evidence of positive selection during acute infection. Previous reports have relied on infections in which selective pressures are likely to be particularly strong (e.g. due to drug treatment or infection with a poorly adapted virus), or in which the virus has replicated within a host for an extended period of time (*Xue et al., 2017*). Under these conditions, it is plausible that positively selected alleles reach levels of detection. We suggest that these are important and informative exceptions to dynamics defined here, in which positive selection is rarely strong enough to drive a new mutation to a frequency above 2% over the course of several days. Our findings are consistent with the observed global rates of influenza evolution and with epidemiological studies that have shown limited antigenic selection over the course of local epidemics (*Nelson et al., 2006*; *Bedford et al., 2015*).

We used both a simple presence-absence model and a more complex beta binomial model to estimate an extremely tight transmission bottleneck. The estimation of a small bottleneck is due to the fact that at most polymorphic sites in the donor, only one allele (iSNV) was found in the recipient. While our methods for variant calling may be more conservative than those used in similar studies, we found that relaxing our variant calling criteria only led to the inclusion of false positive variants and did not significantly inflate our estimates. Furthermore, the beta binomial model accounts for false negative iSNV (i.e. variants that are transmitted but not detected in the donor), which can lead to underestimated transmission bottlenecks (*Sobel Leonard et al., 2017*). Our formulation of this model incorporates empirically determined sensitivity and specificity metrics to

account for both false negative iSNV and false positive iSNV (*McCrone and Lauring, 2016*). Finally, if rare, undetected, iSNV were shared between linked individuals, we would expect to see transmission of more common iSNV (frequency 5–10%), which we can detect with high sensitivity. In our data, the transmission probability iSNVs > 5% frequency in the donor were also well predicted by small bottleneck size (*Figure 3D and E*).

Although the size of our transmission bottleneck is consistent with estimates obtained for other viruses and in experimental animal models of influenza (*Zwart and Elena, 2015*; *Varble et al., 2014*), it differs substantially from the only other study of bottlenecks in natural human infection (*Poon et al., 2016*; *Sobel Leonard et al., 2017*). While there are significant differences in the design and demographics of the cohorts, the influenza seasons under study, and sequencing methodology (*Kugelman et al., 2017*), the bottleneck size estimates are fundamentally driven by the amount of viral diversity shared among individuals in a household. Importantly, and as in Poon et al., we used both epidemiologic linkage and the genetic relatedness of viruses in households to define transmission pairs and to exclude confounding from the observed background diversity in the community. We find that household transmission pairs and randomly assigned community pairs had distinct patterns of shared consensus and minority variant diversity, a pattern that is distinct from the observations of Poon et al. An unexplained aspect of their study is that rare iSNV were frequently shared by randomly selected individuals, and more common ones were not (*Poon et al., 2016*).

Accurately modeling and predicting influenza virus evolution requires a thorough understanding of the virus' population structure. Some models have assumed a large intrahost population and a relatively loose transmission bottleneck (*Geoghegan et al., 2016*; *Russell et al., 2012*; *Peck et al., 2015*). Here, adaptive iSNV can rapidly rise in frequency and low frequency variants can have a high probability of transmission. In such a model, it would be possible for an emerging virus to become more transmissible or a seasonal virus to evolve resistance to vaccine-induced immunity over a short transmission chain (*Herfst et al., 2012*; *Russell et al., 2012*). Although the dynamics of emergent avian influenza and human adapted seasonal viruses likely differ (*Petrova and Russell, 2017*), our work suggests that fixation of multiple mutations over the course of a single acute infection is unlikely.

While it may seem counterintuitive that influenza evolution is dominated by stochasticity on local scales and positive selection on global scales, these models are certainly not in conflict. We have deeply sequenced 332 intrahost populations from 283 individuals collected over more than 11,000 person-seasons of observation and only identified a handful of minority antigenic variants with limited evidence for positive selection (this work and [*Debbink et al., 2017*]). Importantly, our data suggest that even if selection acts below our level of detection, such rare variants are unlikely to transmit. Given the size of the estimated bottleneck, the probability of transmission is approximately 1.7% for a variant at 1% frequency and 3.3% for a variant at 2% frequency. However, with several million infected individuals each year, even inefficient processes and rare events at the scale of individual hosts are likely to occur at a reasonable frequency on a global scale.

## Materials and methods

### Description of the cohort

The HIVE cohort (*Monto et al., 2014*; *Ohmit et al., 2013*, *2015*, *2016*; *Petrie et al., 2013*), established at the UM School of Public Health in 2010, enrolled and followed households of at least three individuals with at least two children < 18 years of age; households were then followed prospectively throughout the year for ascertainment of acute respiratory illnesses. Study participants were queried weekly about the onset of illnesses meeting our standard case definition (two or more of: cough, fever/feverishness, nasal congestion, sore throat, body aches, chills, headache if ≥3 years old; cough, fever/feverishness, nasal congestion/runny nose, trouble breathing, fussiness/irritability, decreased appetite, fatigue in <3 years old), and the symptomatic participants then attended a study visit at the research clinic on site at UM School of Public Health for sample collection. For the 2010–2011 through 2013–2014 seasons, a combined nasal and throat swab (or nasal swab only in children < 3 years of age) was collected at the onsite research clinic by the study team. Beginning with the 2014–2015 seasons, respiratory samples were collected at two time points in each participant meeting the case definition; the first collection was a self- or parent-collected nasal swab collected at illness

onset. Subsequently, a combined nasal and throat swab (or nasal swab only in children < 3 years of age) was collected at the onsite research clinic by the study team. Families with very young children (<3 years of age) were followed using home visits by a trained medical assistant.

Active illness surveillance and sample collection for cases were conducted October through May and fully captured the influenza season in Southeast Michigan in each of the study years. Data on participant, family and household characteristics, and on high-risk conditions were additionally collected by annual interview and review of each participant's electronic medical record. In the current cohort, serum specimens were also collected twice yearly during fall (November-December) and spring (May-June) for serologic testing for antibodies against influenza.

This study was approved by the Institutional Review Board of the University of Michigan Medical School. Adults provided written informed consent for participation for themselves and their children; children 7–17 years provided oral assent.

## Identification of influenza virus

Respiratory specimens were processed daily to determine laboratory-confirmed influenza infection. Viral RNA was extracted (Qiagen QIAamp Viral RNA Mini Kit, Germantown, MD) and tested by RT-PCR for universal detection of influenza A and B. Samples with positive results by the universal assay were then subtyped to determine A(H3N2), A(H1N1), A(pH1N1) subtypes and B(Yamagata) and B(Victoria) lineages. We used primers, probes and amplification parameters developed by the Centers for Disease Control and Prevention Influenza Division for use on the ABI 7500 Fast Real-Time PCR System platform. An RNAseP detection step was run for each specimen to confirm specimen quality and successful RNA extraction.

## Quantification of viral load

Quantitative reverse transcription polymerase chain reaction (RT-qPCR) was performed on 5 µl RNA from each sample using CDC RT-PCR primers InfA Forward, InfA Reverse, and InfA probe, which bind to a portion of the influenza M gene (CDC protocol, 28 April 2009). Each reaction contained 5.4 µl nuclease-free water, 0.5 µl each primer/probe, 0.5 µl SuperScript III RT/Platinum Taq mix (Invitrogen111732, Carlsbad, CA) 12.5 µl PCR Master Mix, 0.1 µl ROX, 5 µl RNA. The PCR master mix was thawed and stored at 4°C, 24 hr before reaction set-up. A standard curve relating copy number to Ct value was generated based on 10-fold dilutions of a control plasmid run in duplicate.

## Illumina library preparation and sequencing

We amplified cDNA corresponding to all eight genomic segments from 5 µl of viral RNA using the SuperScript III One-Step RT-PCR Platinum Taq HiFi Kit (Invitrogen 12574). Reactions consisted of 0.5 µl Superscript III Platinum Taq Mix, 12.5 µl 2x reaction buffer, 6 µl DEPC water, and 0.2 µl of 10 µM Uni12/Inf1, 0.3 µl of 10 µM Uni12/Inf3, and 0.5 µl of 10 µM Uni13/Inf1 universal influenza A primers (*Zhou et al., 2009*). The thermocycler protocol was: 42 °C for 60 min then 94 °C for 2 min then 5 cycles of 94 °C for 30 s, 44 °C for 30 s, 68 °C for 3 min, then 28 cycles of 94 °C for 30 s, 57 °C for 30 s, 68 °C for 3 min. Amplification of all eight segments was confirmed by gel electrophoresis, and 750 ng of each cDNA mixture were sheared to an average size of 300 to 400 bp using a Covaris (Woburn, MA) S220 focused ultrasonicator. Sequencing libraries were prepared using the NEBNext Ultra DNA library prep kit (NEB E7370L), Agencourt AMPure XP beads (Beckman Coulter A63881, Indianapolis, IN), and NEBNext multiplex oligonucleotides for Illumina (NEB E7600S, Ipswich, MA). The final concentration of each barcoded library was determined by Quanti PicoGreen dsDNA quantification (ThermoFisher Scientific, Waltham, MA), and equal nanomolar concentrations were pooled. Residual primer dimers were removed by gel isolation of a 300–500 bp band, which was purified using a GeneJet Gel Extraction Kit (ThermoFisher Scientific). Purified library pools were sequenced on an Illumina HiSeq 2500 with 2 × 125 nucleotide paired end reads. All raw sequence data have been deposited at the NCBI sequence read archive (BioProject Accession number: PRJNA412631). PCR amplicons derived from an equimolar mixture of eight clonal plasmids, each containing a genomic segment of the circulating strain were processed in similar fashion and sequenced on the same HiSeq flow cell as the appropriate patient derived samples. These clonally derived samples served as internal controls to improve the accuracy of variant identification and control for batch effects that confound sequencing experiments.

## Identification of iSNV

Intrahost single nucleotide variants were identified in samples that had greater than $10^3$ genomes/μl and an average coverage >1000 x across the genome. Variants were identified using DeepSNV and scripts available at https://github.com/lauringlab/variant_pipeline as described previously (*McCrone and Lauring, 2016*; copy archived at https://github.com/elifesciences-publications/variant_pipeline) with a few minor and necessary modifications. Briefly, reads were aligned to the reference sequence (H3N2 2010–2011 and 2011–2012: GenBank CY121496-503, H3N2 2012–2013: GenBank KJ942680-8, H3N2 2014–2015: Genbank CY207731-8, H1N1 GenBank: CY121680-8) using Bowtie2 (35). Duplicate reads were then marked and removed using Picard (http://broadinstitute.github.io/picard/). We identified putative iSNV using DeepSNV. Bases with phred <30 were masked. Minority iSNV (frequency <50%) were then filtered for quality using our empirically determined quality thresholds (p-value<0.01 DeepSNV, average mapping quality >30, average Phred > 35, average read position between 31 and 94). To control for RT-PCR errors in samples with lower input titers, all isolates with titers between $10^3$ and $10^5$ genomes/μl were processed and sequenced in duplicate. Only iSNV that were found in both replicates were included in down stream analysis. The frequency of the variant in the replicate with higher coverage at the iSNV location was assigned as the frequency of the iSNV. Finally, any SNV with a frequency below 2% was discarded.

Given the diversity of the circulating strain in a given season, there were a number of cases in which isolates contained mutations that were essentially fixed (>95%) relative to the plasmid control. Often in these cases, the minor allele in the sample matched the major allele in the plasmid control. We were, therefore, unable to use DeepSNV in estimating the base specific error rate at this site for these minor alleles and required an alternative means of discriminating true and false minority iSNV. To this end we applied stringent quality thresholds to these putative iSNV and implemented a 2% frequency threshold. In order to ensure we were not introducing a large number of false positive iSNV into our analysis, we performed the following experiment. Perth (H3N2) samples were sequenced on the same flow cell as both the Perth and Victoria (H3N2) plasmid controls. Minority iSNV were identified using both plasmid controls. This allowed us to identify rare iSNV at positions in which the plasmid controls differed both with and without the error rates provided by DeepSNV. We found that at a frequency threshold of 2% the methods were nearly identical (NPV of 1, and PPV of 0.94 compared to DeepSNV).

## Overview of models used for estimating the transmission bottleneck

We model transmission as a simple binomial sampling process (*Sobel Leonard et al., 2017*). In our first model, we assume any transmitted iSNV, no matter the frequency, will be detected in the recipient. In the second, we relax this assumption and account for false negative iSNV in the recipient. To include the variance in the transmission bottlenecks between pairs we use maximum likelihood optimization to fit the average bottleneck size assuming the distribution follows a zero-truncated Poisson distribution.

## Presence/Absence model

The presence/absence model makes several assumptions. We assume perfect detection of all transmitted iSNV in the recipient. For each donor iSNV, we measure only whether or not the variant is present in the recipient. Any iSNV that is not found in the recipient is assumed to have not been transmitted. We also assume the probability of transmission is determined only by the frequency of the iSNV in the donor at the time of sampling (regardless of how much time passes between sampling and transmission). The probability of transmission is simply the probability that the iSNV is included at least once in a sample size equal to the bottleneck. Finally, we assume all genomic sites are independent of one another. For this reason, we discarded the one case where the donor was likely infected by two strains, as the iSNV were certainly linked.

Because the presence/absence model is unaware of the frequency of alleles in the recipient we must track both alleles at each donor polymorphic site.

Let $A_1$ and $A_2$ be alleles in donor $j$ at genomic site $i$. Let $P(A_1)$ be the probability that $A_1$ is the only transmitted allele. There are three possible outcomes for each site. Either only $A_1$ is transmitted, only $A_2$ is transmitted, or both $A_1$ and $A_2$ are transmitted. The probability of only $A_1$ being transmitted given a bottleneck size of $N_b$ is

$$P_{i,j}(A_1 \mid N_b) = p_1^{N_b} \tag{1}$$

where $p_1$ is the frequency of $A_1$ in the donor. In other words, this is simply the probability of only drawing $A_1$ in $N_b$ draws. The probability that only $A_2$ is transmitted is similarly defined.

The probability of both alleles being transmitted is given by

$$P_{i,j}(A_1, A_2 \mid N_b) = 1 - \left(p_1^{N_b} + p_2^{N_b}\right) \tag{2}$$

where $p_1$ and $p_2$ are the frequencies of the alleles respectively. This is simply the probability of not picking only $A_1$ or only $A_2$ in $N_b$ draws.

This system could easily be extended to cases where there are more than two alleles present at a site; however, that never occurs in our data set.

For ease, we will denote the likelihood of observing the data at a polymorphic site $i$ in each donor $j$ given the bottleneck size $N_b$ as $P_{i,j}(N_b)$ where $P_{i,j}(N_b)$ is defined by equation 1 if only one allele is transmitted and equation 2 if two alleles are transmitted.

The log likelihood of a bottleneck of size $N_b$ is given by

$$LL(N_b) = \sum_j \sum_i \mathrm{Ln}(P_{i,j}) \tag{3}$$

where $i,j$ refers to the $i$th polymorphic site in the $j$th donor. This is the log of the probability of observing the data summed over all polymorphic sites across all donors.

Because the bottleneck size is likely to vary across transmission events, we used maximum likelihood to fit the bottleneck distribution as opposed to fitting a single bottleneck value. Under this model we assumed the bottlenecks were distributed according to a zero-truncated Poisson distribution parameterized by $\lambda$. The likelihood of observing the data given a polymorphic site $i$ in donor $j$ and $\lambda$ is

$$P_{i,j}(\lambda) = \sum_{N_b=1}^{\infty} P_{i,j}(N_b) P(N_b \mid \lambda) \tag{4}$$

where $P_{i,j}(N_b)$ is defined as above, $P(N_b \mid \lambda)$ is the probability of drawing a bottleneck of size $N_b$ from a zero-truncated Poisson distribution with a mean of $\frac{\lambda}{1-e^{-\lambda}}$. The sum is across all possible $N_b$ defined on $[1, \infty)$. Unless otherwise noted, we only investigated bottleneck sizes up to 100, as initial analyses suggested $\lambda$ is quite small and the probability of drawing a bottleneck size of 100 from a zero-truncated Poisson distribution with $\lambda = 10$ is negligible. We follow this convention whenever this sum appears.

The log likelihood of $\lambda$ for the data set is given by

$$LL(\lambda) = \sum_j \sum_i \mathrm{Ln}\left(\sum_{N_b=1}^{\infty} P_{i,j}(N_b) P(N_b \mid \lambda)\right) \tag{5}$$

## Beta binomial model

The Beta binomial model is explained in detail in Leonard *et al.* (Sobel Leonard et al. 2017). It is similar to the presence/absence model in that transmission is modeled as a simple sampling process; however, it relaxes the following assumptions. In this model, the frequencies of transmitted variants are allowed to change between transmission and sampling according a beta distribution. The distribution is not dependent on the amount of time that passes between transmission and sampling, but rather depends on the size of the founding population (here assumed to equal to $N_b$) and the number of variant genomes present in founding population $k$. Note the frequency in the donor is assumed to be the same between sampling and transmission.

The equations below are very similar to those presented by Leonard *et al.* with one exception. Because we know the sensitivity of our method to detect rare variants based on the expected frequency and the titer, we can include the possibility that iSNV are transmitted but are missed due to poor sensitivity. Because the beta binomial model is aware of the frequency of the iSNV in the recipient, no information is added by tracking both alleles at a genomic site $i$.

Let $p_{i,j_d}$ represent the frequency of the minor allele at position $i$ in the donor of some transmission pair $j$. Similarly, let $p_{i,j_r}$ be the frequency of that same allele in the recipient of the $j$th transmission pair. Then, as in Leonard et al., the likelihood of some bottleneck $N_b$ for the data at site $i$ in pair $j$ where the minor allele is transmitted is given by

$$L(N_b)_{i,j} = \sum_{k=1}^{N_b} \text{p\_beta}(p_{i,j_r} \mid k, N_b - k) \, \text{p\_bin}(k \mid N_b, p_{i,j_d}) \tag{6}$$

Where p_beta is the probability density function for the beta distribution and p_bin is the probability mass function for the binomial distribution.

This is the probability density that the transmitted allele is found in the recipient at a frequency of $p_{i,j_r}$ given that the variant was in $k$ genomes in a founding population of size $N_b$ times the probability of drawing $k$ variant genomes in a sample size of $N_b$ and a variant frequency of $p_{i,j_d}$. This is then summed for all possible $k$ where $1 \leq k \leq N_b$.

As in equation 4 the likelihood of a zero truncated Poisson with a mean of $\frac{\lambda}{1 - e^{-\lambda}}$ given this transmitted variants is then given by

$$L(\lambda)_{i,j}^{\text{transmitted}} = \sum_{N_b=1}^{\infty} L(N_b)_{i,j} P(N_b \mid \lambda) \tag{7}$$

This is simply the likelihood of each $N_b$ weighted by the probability of drawing a bottleneck size of $N_b$ from bottleneck distribution.

In this model, there are three possible mechanisms for a donor iSNV to not be detected in the recipient. (i) The variant was not transmitted. (ii) The variant was transmitted but is present below our limit of detection (2%). (iii) The variant was transmitted and is present above our limit of detection but represents a false negative in iSNV identification.

As in Leonard et al., the likelihood of scenarios (i) and (ii) for a given $N_b$ are expressed as

$$L(N_b)_{i,j}^{\text{lost}} = \sum_{k=0}^{N_b} \text{p\_beta\_cdf}(p_{i,j_r} < 0.02 \mid k, N_b - k) \, \text{p\_bin}(k \mid N_b, p_{i,j_d}) \tag{8}$$

Where p_beta_cdf is the cumulative distribution function for the beta distribution. Note that if $k = 0$ (i.e. the iSNV was not transmitted) then the term reduces to the probability of not drawing the variant in $N_b$ draws.

The likelihood of the variant being transmitted but not detected in the recipient given a bottleneck of $N_b$ is described by

$$L(N_b)_{i,j}^{\text{missed}} = \sum_{k=0}^{N_b} \sum_{f_e}^{[0.02, 0.05, 0.1]} \text{p\_beta\_cdf}(f_e < p_{i,j_r} < f_{e+1} \mid k, N_b - k) \times$$
$$\text{p\_bin}(k \mid N_b, p_{i,j_d})(\text{FNR} \mid \text{Titer}_r, f_e) \tag{9}$$

This is the likelihood of the variant existing in the ranges [0.02,0.05] or [0.05,0.1] given an initial frequency of $k/N_b$ and a bottleneck size of $N_b$ multiplied by the expected false negative rate (FNR) given the titer of the recipient and the lower frequency bound. We assumed perfect sensitivity for detection of iSNV present above 10%, rounded recipient titers down to the nearest $\log_{10}$ titer (e.g. $10^3, 10^4, 10^5$) and assumed the entire range $[f_e, f_{e+1}]$ has the same sensitivity as the lower bound.

The likelihood of $\lambda$ for iSNV that are not observed in the recipient is then given by summing equations 8 and 9 across all possible $N_b$.

$$L(\lambda)_{i,j}^{\text{nontransmitted}} = \sum_{N_b=1}^{\infty} \left( L(N_b)_{i,j}^{\text{lost}} + L(N_b)_{i,j}^{\text{missed}} \right) P(N_b \mid \lambda) \tag{10}$$

The log likelihood of the total dataset is then determined by summing the log of equations 7 and 10 (as applicable) across all polymorphic sites in each donor. (As before here we sum of $N_b$ within the range $[1, 100]$.)

## Model comparisons

In order evaluate the fits of the two transmission models, we calculated the probability of detecting a donor iSNV in the recipient given its frequency in the donor. For each model this reduces to one minus the probability of only detecting the other allele in the recipient summed over all possible bottlenecks. As in fitting the models this infinite sum was approximated by a partial sum up to a bottleneck size of 100. We used a recipient titer of $10^4$ genomes/µl to estimate the effect of sequencing sensitivity for the beta binomial model.

Annotated computer code for all analyses and for generating the figures can be accessed at https://github.com/lauringlab/Host_level_IAV_evolution. (copy archived at https://github.com/elifes-ciences-publications/Host_level_IAV_evolution)

## Acknowledgements

This work was supported by a Clinician Scientist Development Award from the Doris Duke Charitable Foundation (CSDA 2013105), a University of Michigan Discovery Grant, and R01 AI118886, all to ASL. The HIVE cohort was supported by NIH R01 AI097150 and CDC U01 IP00474 to ASM. JTM was supported by the Michigan Predoctoral Training Program in Genetics (T32GM007544). RJW was supported by K08 AI119182. We thank Katherine Xue for suggestions on analysis of globally circulating variants and Alexey Kondrashov and Aaron King for helpful discussion.

## Additional information

### Funding

| Funder | Grant reference number | Author |
| --- | --- | --- |
| National Institute of General Medical Sciences | T32 GM007544 | John T McCrone |
| National Institute of Allergy and Infectious Diseases | K08 AI119182 | Robert J Woods |
| Centers for Disease Control and Prevention | U01 IP00474 | Arnold S Monto |
| National Institute of Allergy and Infectious Diseases | R01 AI097150 | Arnold S Monto |
| Doris Duke Charitable Foundation | CSDA 2013105 | Adam S Lauring |
| National Institute of Allergy and Infectious Diseases | R01 AI118886 | Adam S Lauring |
| University of Michigan | Discovery grant | Adam S Lauring |

The funders had no role in study design, data collection and interpretation, or the decision to submit the work for publication.

### Author contributions

John T McCrone, Resources, Data curation, Software, Formal analysis, Investigation, Visualization, Methodology, Writing—original draft, Writing—review and editing; Robert J Woods, Formal analysis, Investigation, Writing—review and editing; Emily T Martin, Resources, Data curation, Project administration, Writing—review and editing; Ryan E Malosh, Data curation, Writing—review and editing; Arnold S Monto, Data curation, Investigation, Methodology, Project administration, Writing—review and editing; Adam S Lauring, Conceptualization, Data curation, Formal analysis, Supervision, Funding acquisition, Investigation, Methodology, Writing—original draft, Project administration, Writing—review and editing

Author ORCIDs
John T McCrone (iD) http://orcid.org/0000-0002-9846-8917
Ryan E Malosh (iD) http://orcid.org/0000-0003-3546-5935
Adam S Lauring (iD) http://orcid.org/0000-0003-2906-8335

Ethics

Human subjects: This study was approved by the Institutional Review Board of the University of Michigan Medical School. Adults provided written informed consent for participation for themselves and their children; children 7-17 years provided oral assent.

Decision letter and Author response

Decision letter https://doi.org/10.7554/eLife.35962.035
Author response https://doi.org/10.7554/eLife.35962.036

## Additional files

### Supplementary files

• Supplementary file 1. Sensitivity and specificity of variant detection
DOI: https://doi.org/10.7554/eLife.35962.028

• Supplementary file 2. Nonsynonymous substitutions in HA antigenic sites
DOI: https://doi.org/10.7554/eLife.35962.029

• Supplementary file 3. Estimated bottleneck size for individual transmission pairs
DOI: https://doi.org/10.7554/eLife.35962.030

• Transparent reporting form
DOI: https://doi.org/10.7554/eLife.35962.031

### Data availability

All data generated or analyzed during this study are included in the manuscript and supporting files. Source data files have been provided. All sequence reads have been deposited to NCBI's BioProject under accession number PRJNA412631.

The following dataset was generated:

| Author(s) | Year | Dataset title | Dataset URL | Database, license, and accessibility information |
|---|---|---|---|---|
| John T McCrone, Robert J Woods, Emily T Martin, Ryan E Malosh, Arnold S Monto, Adam S Lauring | 2018 | Whole genome sequencing of Influenza isolates from a prospective household cohort | https://www.ncbi.nlm.nih.gov/bioproject/PRJNA412631 | Publicly available at the NCBI BioProject (accession no: PRJNA412631). |

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
