## [Decision Letter]

[Editors’ note: a previous version of this study was rejected after peer review, but the authors submitted for reconsideration. The first decision letter after peer review is shown below.]

Thank you for submitting your work entitled "Stochastic processes dominate the within and between host evolution of influenza virus" for consideration by *eLife*. Your article has been reviewed by three peer reviewers, one of whom, Richard A Neher, is a member of our Board of Reviewing Editors, and the evaluation has been overseen by a Senior Editor. The following individuals involved in review of your submission have agreed to reveal their identity: Katia Koelle (Reviewer #2); Christopher J R Illingworth (Reviewer #3).

Our decision has been reached after consultation between the reviewers. Based on these discussions and the individual reviews below, we regret to inform you that this manuscript will not be considered further for publication in *eLife*. As detailed below, the reviewers were not convinced by the estimation of within-host Ne but unanimously felt the characterization of the transmission bottleneck is interesting and robust. We, therefore, encourage you to resubmit a manuscript focusing on the transmission bottleneck as a short report.

All reviewers agreed that you have presented comprehensive data on within-host diversity of influenza virus. The data set includes transmission pairs within households which allowed you to estimate the size of the transmission bottleneck. In sharp contrast to previous estimates, your data suggest a very small transmission bottleneck (evident in Figure 3C). This observation is important, and this conclusion is strongly supported by the data. The estimation of Ne, however, is unconvincing. The major concerns regarding the Ne estimation are summarized below -- you can find details in the attached reviews.

1) The signal to estimate Ne is weak. The distribution of iSNV frequency changes as day 3 is hardly different from that at day 1. Furthermore, the distribution of variant frequency changes is long-tailed, which conflicts the assumptions of the Wright-Fisher model. While you correct for false negatives and allow for a constant variance in observed frequency, estimates might still be affected by extraction and amplification noise (particularly important for the second time point with typically lower viral load). At best, your estimate of Ne is a loose lower bound.

2) Whether Ne is a well-defined quantity of an acute infection is debatable. The populations you sequenced may be a mixture of different subpopulations from different patches of infected tissue. Changes in variant frequencies of common iSNVs are therefore might not due to resampling noise, but differential growth and shrinkage of subpopulations. This type of noise is not characterized by an effective population size.

3) Your estimate of the mutation rate (3-4e-5/site/replication) is inconsistent with the 5fold higher rate you estimated in Pauly et al., 2017, despite generous assumptions. How can this difference be reconciled?

Reviewer #1:

McCrone et al., study genetic variation of influenza virus population within hosts and in transmission chain to estimate the stochasticity of within host evolution and the effective size of the transmission bottle neck. Virus populations are characterized by deep sequencing. The quality of the data and the analysis seem high and the conclusions seem justified and solid. I have a number of comments regarding the interpretation of the results and some parts of the analysis.

While the transmission bottleneck size is a relevant and interpretable quantity, I don't think the concept of an effective population is useful for an acute within-host population. The actual population size is orders of magnitudes higher than any estimate of Ne and the variation in variant frequencies likely depends on subpopulations dynamics in the infected tissue. These are populations subject to exponential expansion and decay and there is no reason why a diffusion-based model is useful. This is evident in Figure 2B: The distribution of allele frequency changes is incompatible with a diffusion model: most alleles don't change much in frequency, while others change by 0.4. The removal of the most extremely changing iSNVs partly addresses this issue, but also shows that there is a substantial dependence on this cut-off. The distribution of variant frequency changes is interesting in its own right, but there is probably not enough data to investigate this here. But even taking the estimate of Ne at face value, how is sampling noise handled in the Ne estimates? And why is the 0.02 cutoff a good idea for the later sample? Once a iSNV is ascertained in the earlier sample, you should use the raw variant frequency in the diffusion model.

While fitting a diffusion model is not appropriate, one could still estimate how a quantity like the inter quartile range of (x_1-x_0)/(x_0(1-x_0)) increases with time. A quick and dirty analysis using the data the authors uploaded as supporting information suggests that there is little signal.

The estimate of selection coefficients is not terribly convincing. It seems much more likely that from one day to another, different patches of infected tissue contribute to different degree to the sample and such population shifts will result in changing frequencies.

The estimates of the size of transmission bottleneck, on the other hand, are much better defined and a more relevant quantity than the within host Ne. The estimate seems solid and provide an interesting counterpoint to high estimates from previous studies. It would be interesting to investigate whether child-child, parent-child, and adult-adult transmission have distinct properties.

I find the statements that the potential for spread of a mutation is determined by the transmission bottleneck are misleading. Mutations that rise in frequency within a host have a much higher frequency to be transmitted than other mutations, even if they rise only to frequencies of 1/1000 or less.

To assess the reliability of the variant calling and diversity representation, I would like to see the analogs of Figure 2—figure supplement 1A for samples that were processed in duplicate.

A more systematic investigation of whether antigenic sites tend to segregate within individuals seems possible and important. Why not compare distributions of variant frequencies at sites with global variation in the respective season with those that are globally monomorphic? Similar comparisons could be done for epitope vs non-epitope, or syn vs non-syn.

Any idea why the probability to transmit a 50% variant is larger than 50%? Is this the effect of minor variants and bottlenecks>1 or selective advantage of minor variants? The explanation of the shaded areas is confusing: These are distributions of the inferred probability across 1000 simulations, but not outcomes of transmission.

The method to estimate the mutation rate is problematic. First of all, Eq. 23 is incorrect as written and the model assumptions are unlikely to hold in an infected individual.

Reviewer #2:

In this manuscript, McCrone and coauthors use deep sequencing data from a prospective community-based cohort study to estimate the transmission bottleneck size for seasonal influenza viruses H1N1 and H3N2, as well as the effective population size of the viral population within acutely infected individuals. In contrast to the only other dataset used in quantifying transmission bottleneck sizes for influenza circulating in natural human populations (Poon et al.), they find evidence for a remarkably small bottleneck size of 1-2 influenza virions using two distinct methods. Further, they estimate small within-host effective population sizes of 30-40 virions using a number of different methods. With both small transmission bottleneck sizes and small within-host population sizes, they conclude that genetic drift and stochastic processes are important factors influencing influenza virus dynamics. Finally, they do find some evidence for purifying selection going on within infected hosts, given observed patterns of nonsynonymous versus synonymous nucleotide variation.

In general, this manuscript is very clearly written, is very thorough in applying different methods to arrive at robust conclusions and presents interesting results. One concern is of course the disparity between the results this manuscript presents for transmission bottleneck sizes in flu (1-2 virions) relative to the previous literature estimate of 100-200 virions, based on the data presented in Poon et al. It is clear from the variant frequency plot shown in Figure 3C that these differences in estimated bottleneck sizes are based on differences present in the data themselves, rather than the specifics of the methods applied to the data.

I have several concerns, but none that are sufficiently major to keep this paper from being considered for publication in *eLife*.

If the within-host effective population size is very small (~35 virions), then it seems to me that the timing of transmission should matter for determining the transmission bottleneck size. Have you looked to see how much the estimates differ from one another based on whether the first or second sampling timepoint in a donor was used? (The data might not be available to conduct this analysis…)

Subsection “Within host populations have low genetic diversity”: I really like that the within-host Ne calculation was done a number of different ways and that similar results were obtained when some of the more stringent assumptions (site independence and large population size) of the first approach were relaxed. One other assumption in all of these models, however, I think is that the viral population size is constant. Given that this is an acute infection, the viral population size is much more likely to be exponentially declining (given that individuals were sampled following symptom onset). How does an exponentially declining viral population affect your estimate of within-host Ne?

Subsection “The mutation rate of influenza A virus within human hosts”: here, you estimate a mutation rate on the order of 3-4 x 10^-5 mutations per site per replication cycle, which corresponds to ~0.35 mutations/genome/replication cycle. You also mention that this is close to your own recently published estimate (Pauly et al., 2017). However, that paper estimated a mutation rate on the order of 1.8 x 10^-4 (H1N1) and 2.5 x 10^-4, resulting in 2-3 mutations/genome per replication cycle. Since that Pauly et al., paper's primary point was to revise the 2.7 x 10^-6 – 3.0 x 10^-5 previous estimates, it would be nice to explicitly state the discrepancy of the Pauly et al., estimate with the estimate obtained in this paper (since they differ by an order of magnitude).

*Reviewer #3:*

This work presents an interesting dataset collected from a household study of human influenza infection. Via a mathematical analysis, two key claims about the data are presented here. Firstly, the within-host effective population size, Ne, is relatively low, at 30-70. Secondly the transmission bottleneck is tight, with 1-2 viruses typically founding an infection.

There is an odd repetition to the manuscript, in so far as multiple techniques are applied to get each value of N. In so far as a method of analysis is correct, recalculation of the same sum using an alternative approach is unnecessary. In so far as a method has limitations, repeating a calculation using another method, which shares the same limitations, adds no further useful information to the manuscript. Quantitative accuracy is not achieved via a democratic vote among statistical methods.

The analysis of the transmission bottleneck is convincing, though susceptible to criticism under the point above. Given that the presence-absence model underestimates the size of a bottleneck, and the conclusion that the bottleneck is small, why not just use the β-binomial model? For myself, under the assumption that the identification of transmission pairs is correct, Figure 3C is clear evidence of a tight bottleneck.

The analysis of within-host effective population size is problematic. The initial analysis of Ne assumes that selection does not act upon the viral populations, calculating the parameter on the basis of a model of genetic drift alone. This is fine in establishing a lower bound for Ne but cannot support a conclusion that Ne is small. Here, while much care is taken in evaluating both diffusion and Wright-Fisher models, and evaluating the potential power of the calculation, the issues addressed are relatively trifling in the context of the assumption of neutrality. The final result is to conclude that Ne is small, and therefore selection has little influence, all under the assumption that selection can be neglected.

An attempt to fix this problem is made via the use of a method which estimates Ne along with a selection coefficient for each SNP. I am not convinced that this method is effective in the context of the data collected. Whereas the original application of this method was to a dataset for which sequence data was collected at 13 separate time-points (Foll et al., 2014), here only two time-points are available for each patient (Results section). In estimating the value of Ne from time-resolved data, the value of selection, speaking loosely, is estimated from the increase or decrease of an allele frequency over time, while the extent of drift (or Ne) is estimated from the extent of deviation of the data from a deterministic model of selection (see e.g. Feder et al., 2014). Where data is collected for only two time-points, a deterministic model (with effectively infinite Ne) can be fitted perfectly to any allele frequency data, with a different selection coefficient being fitted to each variant. I am therefore unclear where the value of Ne estimated from this method arises from; perhaps fitting a prior to the selection coefficient affects this? I note that the method, while validated for the inference of selection from two time-points, is not validated in the original publication for its ability to infer Ne.

There are other factors which might limit the correct inference of a high Ne. For example, while the existence of variant allele frequencies is well-validated, the precision with which an allele frequency can be measured is less clear. A standard deviation in a variant frequency is cited from a previous paper (subsection “Discrete Wright-Fisher estimation of 𝑁𝑒”), albeit this was measured from in vitro material. Other authors (Lakdawala et al., 2015) have highlighted the potential for non-trivial population structure within a host. Here, while one sample was collected via nasal swab, the second was collected from a mix of nasal swab and throat samples; this could conceivably introduce a greater variance into the allele frequencies. Further, if a constant standard deviation is measured from low-frequency variant calls, this might underestimate the variance in higher-frequency calls, which under a binomial model would be frequency-dependent.

In short, while there are excellent parts of this paper, there are also significant problems. Although I am happy with the lower bound inferred for within-host Ne, I am not convinced what can be said about the upper bound for this parameter. The conclusion of a low within-host effective population size may not be valid.

[Editors’ note: what now follows is the decision letter after the authors submitted for further consideration.]

Thank you for submitting your article "Stochastic processes dominate the within and between host evolution of influenza virus" for consideration by *eLife*. Your article has been reviewed by three peer reviewers, one of whom, Richard A Neher, is a member of our Board of Reviewing Editors and the evaluation has been overseen by Arup Chakraborty as the Senior Editor. The following individuals involved in review of your submission have agreed to reveal their identity: Katia Koelle (Reviewer #2); Christopher J R Illingworth (Reviewer #3).

The reviewers have discussed the reviews with one another and the Reviewing Editor has drafted this decision to help you prepare a revised submission.

Summary:

McCrone et al., present influenza A virus whole genome deep sequencing data from 200 infected individuals, including 49 likely transmission pairs. The authors find little within-host genetic diversity. Furthermore, low overlap in genetic diversity between samples in a transmission pair suggests that new infections are typically dominated by one founder virus, i.e., the transmission bottleneck is very small. Since the size of the transmission bottleneck is an important determinant of influenza virus transmission and previous studies estimated a much larger bottleneck, the results reported here are an important contribution.

Essential revisions:

This paper is a resubmission of a previously rejected version and was reviewed by the same reviewers. All reviewers agreed (as before) that the results on the transmission bottleneck are important, but two major concerns remain.

1) Limited power to characterize within-host selection. Since most infections seem to be dominated by a single founder virus, the power to detect within-host selection during a few days of observation is very low. Selection would need to amplify a novel beneficial variant (produced at rates of about 1e-5 per day) to above 1% within 3-4 days in order to be detected in this study. Given the difficulty to call variants below 2%, even strong positive selection (s~10%) remains undetectable. Despite this lack of power, the authors make strong claims about the absence of positive selection and often make semi-quantitative statements without a proper comparison to an expectation.

a) Subsection “Mutations in antigenic epitopes are rare”. While you claim stochastic processes dominate within host and positive selection is inefficient, we don't think you have sufficient evidence for this statement. What you could conclude is "positive selection rarely amplifies a beneficial de novo variant above the limit of detection at 2%".

b) In subsection “Within-host populations have low genetic diversity”, you are comparing R^2 values of variant frequencies of same day samples with the fraction of minor variants recovered in later samples. But an R^2 value and the fraction recovered iSNV are not comparable. To make a convincing case that your sequencing method accurately measures iSNV frequency and that the observed frequency changes are biological we would like to see the equivalent of Figure 2—figure supplement 1A for duplicate samples, ideally stratified by viral load (in subsection “Illumina library preparation and sequencing”, you write that samples were processed in duplicate). Such a control is particularly important for the comparison of iSNV frequencies in longitudinal samples, as the samples that are being compared likely differ in viral load. Figure 2—figure supplement 1A could use a 1:1 line. It seems that iSNV frequencies are systematically lower in-home isolates than in clinic isolates for low-frequency variants.

c) Given the difficulty to characterize within host stochasticity and the absence of positive selection, the claims made in this direction need to be revisited, toned down or removed. Furthermore, we suggest changing the title. A title like "Small transmission bottlenecks and low genetic diversity in acute influenza A virus infection". We are open to suggestions here, but please make sure that the title reflects what can be concluded from your data. On larger spatio-temporal scales, influenza virus evolution is certainly not stochastic and the current title doesn't make this distinction clearly enough.

2) Confidence intervals on the transmission bottleneck size. You estimate the parameter of a Poisson distribution that is assumed to describe the distribution of bottlenecks in the population. The error bars of this estimate are remarkably tight given the small number of informative iSNVs in individual pairs. The actual distribution of bottleneck sizes in the population is certainly much broader and fitting a Poisson distribution will underestimate the confidence interval. Please include the 95% confidence intervals for the estimate of each transmission pair in Supplementary file 3. Aggregating these confidence intervals or fitting a broader distribution (say log-normal or power-law) should result in more sensible confidence intervals.

3) Dependence of bottleneck estimates on sample pairings. We suggest, to estimate, bottlenecks from all possible different sample-pairings of donor and recipient in those transmission pairs where either has more than one sample available. In particular the difference between self-sampling and sampling in the clinics (Figure 2—figure supplement 1A suggests a possible systematic deviation) should be tested and discussed.

We suggest resubmitting this manuscript as a short report focusing on transmission bottlenecks. Figure 4 is a plausible supplement to Figure 3.

---

## [Author Response]

[Editors’ note: the author responses to the first round of peer review follow.]

Reviewer #1:McCrone et al., study genetic variation of influenza virus population within hosts and in transmission chain to estimate the stochasticity of within host evolution and the effective size of the transmission bottle neck. Virus populations are characterized by deep sequencing. The quality of the data and the analysis seem high and the conclusions seem justified and solid. I have a number of comments regarding the interpretation of the results and some parts of the analysis.While the transmission bottleneck size is a relevant and interpretable quantity, I don't think the concept of an effective population is useful for an acute within-host population. The actual population size is orders of magnitudes higher than any estimate of Ne and the variation in variant frequencies likely depends on subpopulations dynamics in the infected tissue. These are populations subject to exponential expansion and decay and there is no reason why a diffusion-based model is useful. This is evident in Figure 2B: The distribution of allele frequency changes is incompatible with a diffusion model: most alleles don't change much in frequency, while others change by 0.4. The removal of the most extremely changing iSNVs partly addresses this issue, but also shows that there is a substantial dependence on this cut-off. The distribution of variant frequency changes is interesting in its own right, but there is probably not enough data to investigate this here. But even taking the estimate of Ne at face value, how is sampling noise handled in the Ne estimates? And why is the 0.02 cutoff a good idea for the later sample? Once a iSNV is ascertained in the earlier sample, you should use the raw variant frequency in the diffusion model.While fitting a diffusion model is not appropriate, one could still estimate how a quantity like the inter quartile range of (x_1-x_0)/(x_0(1-x_0)) increases with time. A quick and dirty analysis using the data the authors uploaded as supporting information suggests that there is little signal.

We have removed the diffusion models and all quantitative analysis of effective population size. We have kept the within host data (which the reviewers agreed was of high quality) and discuss it in a qualitative or semi-quantitative manner. We now describe our data as consistent with a model in which variants more frequently arise within hosts through neutral processes such as genetic drift.

The estimate of selection coefficients is not terribly convincing. It seems much more likely that from one day to another different patches of infected tissue contribute to different degree to the sample and such population shifts will result in changing frequencies.

The issue of sampling in a community-based cohort with an acute respiratory viral infection is indeed a challenging one. We note that we provided data in Figure 2—figure supplement 1A of the original manuscript that showed reproducibility of sequence data from two distinct same day samples (one self-collected at home and one collected at the clinic) from six individuals. The reviewer is correct that this does not completely exclude the potential issues of sampling. We have therefore removed the figure and discussion of individual selection coefficients.

The estimates of the size of transmission bottleneck, on the other hand, are much better defined and a more relevant quantity than the within host Ne. The estimate seems solid and provide an interesting counterpoint to high estimates from previous studies. It would be interesting to investigate whether child-child, parent-child, and adult-adult transmission have distinct properties.

We have now provided a Supplementary file 3 that shows the ages and dates of sampling of donor and recipient, bottleneck size, viral subtype, and vaccination status for each transmission pair. We have also generated a shiny app with a link provided (subsection “Identification of forty-three transmission pairs”) so that interested readers can look at the sequence variants observed in each donor and recipient pair.

I find the statements that the potential for spread of a mutation is determined by the transmission bottleneck are misleading. Mutations that rise in frequency within a host have a much higher frequency to be transmitted than other mutations, even if they rise only to frequencies of 1/1000 or less.

If we understand the comment correctly, the reviewer suggests that mutations that increase in frequency within a host have a competitive advantage during transmission, such that a minority variant could be selectively transmitted. This would seem to assume (i) that variants increase in frequency in a host due to positive selection, (ii) that transmission is a selective process as well, (iii) that the same factors are selected within hosts and between hosts. This is a reasonable model, but not well supported by our data. Nevertheless, we have toned down the statement a bit given the uncertainty, and also given that we may misunderstand the comment as stated here. The Introduction now reads. “Their potential for subsequent spread through host populations is heavily dependent on the size of the transmission bottleneck (Alizon et al., 2011; Zwart and Elena, 2015).”

To assess the reliability of the variant calling and diversity representation, I would like to see the analogs of Figure S6 for samples that were processed in duplicate.

The comment seems to suggest that the data presented in Figure 2—figure supplement 1A and elsewhere were based on only one of two samples processed in duplicate. The data are from serial samples (on the same day separated by time) in which variants were ascertained based on duplicate RT-PCR and sequencing libraries for each of the two samples.

As described in the Materials and methods section of the original submission, “To control for PCR errors in samples with lower input titers, all isolates with titers between 10^3^ and 10^5^ genomes/μl were processed and sequenced in duplicate. Only iSNV that were found in both replicates were included in downstream analysis. The frequency of the variant in the replicate with higher coverage at the iSNV location was assigned as the frequency of the iSNV. Finally, any SNV with a frequency below 2% was discarded.”

Therefore, for each sample (nasal/throat swab in 1ml transport media), we performed a single RNA prep. The RNA genome copy number was determined by RT-qPCR. The vast majority of samples had copy numbers between 10^3^ and 10^5^ copies per microliter and were then processed for duplicate multiplex RTPCR reactions and then duplicate Illumina sequencing libraries. We only called iSNV that were present in both replicates subject to our variant calling criteria. We have documented the sensitivity and specificity of this approach in McCrone and Lauring, 2016, used these same criteria in Debbink et al., 2017, and report our validation again in the present manuscript. In our experience, this is well beyond what is often reported in the literature and should suffice for documentation of reliability in variant calling. All figures were based on variants identified using this approach.

A more systematic investigation of whether antigenic sites tend to segregate within individuals seems possible and important. Why not compare distributions of variant frequencies at sites with global variation in the respective season with those that are globally monomorphic? Similar comparisons could be done for epitope vs non-epitope, or syn vs non-syn.

We have now provided a new Figure 2B that shows the distribution of variant frequencies for antigenic and nonantigenic sites in HA (the antigenic sites being the ones likely to be under strongest positive selection). We have also expanded our Figure 2C, which looks at the few within host variants that are found at a global level.

Any idea why the probability to transmit a 50% variant is larger than 50%? Is this the effect of minor variants and bottlenecks>1 or selective advantage of minor variants? The explanation of the shaded areas is confusing: These are distributions of the inferred probability across 1000 simulations, but not outcomes of transmission.

We think the more likely explanation is that it is the effect of minor variants and bottlenecks greater than one, as the reviewer suggests. We observed no convincing evidence of a selective advantage for minor variants across the dataset.

The method to estimate the mutation rate is problematic. First of all, Eq. 23 is incorrect as written and the model assumptions are unlikely to hold in an infected individual.

This section has been removed.

Reviewer #2:In this manuscript, McCrone and coauthors use deep sequencing data from a prospective community-based cohort study to estimate the transmission bottleneck size for seasonal influenza viruses H1N1 and H3N2, as well as the effective population size of the viral population within acutely infected individuals. In contrast to the only other dataset used in quantifying transmission bottleneck sizes for influenza circulating in natural human populations (Poon et al.), they find evidence for a remarkably small bottleneck size of 1-2 influenza virions using two distinct methods. Further, they estimate small within-host effective population sizes of 30-40 virions using a number of different methods. With both small transmission bottleneck sizes and small within-host population sizes, they conclude that genetic drift and stochastic processes are important factors influencing influenza virus dynamics. Finally, they do find some evidence for purifying selection going on within infected hosts, given observed patterns of nonsynonymous versus synonymous nucleotide variation.In general, this manuscript is very clearly written, is very thorough in applying different methods to arrive at robust conclusions and presents interesting results. One concern is of course the disparity between the results this manuscript presents for transmission bottleneck sizes in flu (1-2 virions) relative to the previous literature estimate of 100-200 virions, based on the data presented in Poon et al. It is clear from the variant frequency plot shown in Figure 3C that these differences in estimated bottleneck sizes are based on differences present in the data themselves, rather than the specifics of the methods applied to the data.I have several concerns, but none that are sufficiently major to keep this paper from being considered for publication in eLife.If the within-host effective population size is very small (~35 virions), then it seems to me that the timing of transmission should matter for determining the transmission bottleneck size. Have you looked to see how much the estimates differ from one another based on whether the first or second sampling timepoint in a donor was used? (The data might not be available to conduct this analysis…)

For samples with two within host samples (2014-2015 season only), we used the time point closest to the time of transmission, as determined by date of onset of symptoms in the recipient. We assume that using this sample would provide a better representation of the diversity in the donor at the time of transmission. As above, we have provided the requested information as part of a shiny app with a link (subsection “Identification of forty-three transmission pairs”) that allows one to examine the variants for each pair across samples.

Subsection “Within host populations have low genetic diversity”: I really like that the within-host Ne calculation was done a number of different ways and that similar results were obtained when some of the more stringent assumptions (site independence and large population size) of the first approach were relaxed. One other assumption in all of these models, however, I think is that the viral population size is constant. Given that this is an acute infection, the viral population size is much more likely to be exponentially declining (given that individuals were sampled following symptom onset). How does an exponentially declining viral population affect your estimate of within-host Ne?

As above, we have removed the Ne analysis.

Subsection “The mutation rate of influenza A virus within human hosts”: here, you estimate a mutation rate on the order of 3-4 x 10^-5 mutations per site per replication cycle, which corresponds to ~0.35 mutations/genome/replication cycle. You also mention that this is close to your own recently published estimate (Pauly et al., 2017). However, that paper estimated a mutation rate on the order of 1.8 x 10^-4 (H1N1) and 2.5 x 10^-4, resulting in 2-3 mutations/genome per replication cycle. Since that Pauly et al., paper's primary point was to revise the 2.7 x 10^-6 – 3.0 x 10^-5 previous estimates, it would be nice to explicitly state the discrepancy of the Pauly et al., estimate with the estimate obtained in this paper (since they differ by an order of magnitude).

As above, we have removed this joint analysis of Ne and mutation rate.

Reviewer #3:This work presents an interesting dataset collected from a household study of human influenza infection. Via a mathematical analysis, two key claims about the data are presented here. Firstly, the within-host effective population size, Ne, is relatively low, at 30-70. Secondly the transmission bottleneck is tight, with 1-2 viruses typically founding an infection.There is an odd repetition to the manuscript, in so far as multiple techniques are applied to get each value of N. In so far as a method of analysis is correct, recalculation of the same sum using an alternative approach is unnecessary. In so far as a method has limitations, repeating a calculation using another method, which shares the same limitations, adds no further useful information to the manuscript. Quantitative accuracy is not achieved via a democratic vote among statistical methods.The analysis of the transmission bottleneck is convincing, though susceptible to criticism under the point above. Given that the presence-absence model underestimates the size of a bottleneck, and the conclusion that the bottleneck is small, why not just use the β-binomial model? For myself, under the assumption that the identification of transmission pairs is correct, Figure 3C is clear evidence of a tight bottleneck.

There is no consensus on the right way to analyze a bottleneck. We performed both, not because we pursued a “democratic vote,” but as a way to ensure that our estimates are robust to the assumptions of each model. We are aware of issues regarding presence absence vs. β binomial. The presence-absence model actually provided a marginally better fit for our data. We suspect that if we didn’t present the β binomial, a reviewer would ask that we analyze our data under that model (given the time between transmission and sampling).

The analysis of within-host effective population size is problematic. The initial analysis of Ne assumes that selection does not act upon the viral populations, calculating the parameter on the basis of a model of genetic drift alone. This is fine in establishing a lower bound for Ne but cannot support a conclusion that Ne is small. Here, while much care is taken in evaluating both diffusion and Wright-Fisher models, and evaluating the potential power of the calculation, the issues addressed are relatively trifling in the context of the assumption of neutrality. The final result is to conclude that Ne is small, and therefore selection has little influence, all under the assumption that selection can be neglected.

We have removed analyses related to Ne.

An attempt to fix this problem is made via the use of a method which estimates Ne along with a selection coefficient for each SNP. I am not convinced that this method is effective in the context of the data collected. Whereas the original application of this method was to a dataset for which sequence data was collected at 13 separate time-points (Foll et al., 2014), here only two time-points are available for each patient (Results section). In estimating the value of Ne from time-resolved data, the value of selection, speaking loosely, is estimated from the increase or decrease of an allele frequency over time, while the extent of drift (or Ne) is estimated from the extent of deviation of the data from a deterministic model of selection (see e.g. Feder et al., 2014). Where data is collected for only two time-points, a deterministic model (with effectively infinite Ne) can be fitted perfectly to any allele frequency data, with a different selection coefficient being fitted to each variant. I am therefore unclear where the value of Ne estimated from this method arises from; perhaps fitting a prior to the selection coefficient affects this? I note that the method, while validated for the inference of selection from two time-points, is not validated in the original publication for its ability to infer Ne.

We have removed this analysis, which was related to Ne.

*There are other factors which might limit the correct inference of a high Ne. For example, while the existence of variant allele frequencies is well-validated, the precision with which an allele frequency can be measured is less clear. A standard deviation in a variant frequency is cited from a previous paper (subsection “Discrete Wright-Fisher estimation of 𝑁𝑒”), albeit this was measured from* in vitro *material. Other authors (Lakdawala et al., 2015) have highlighted the potential for non-trivial population structure within a host. Here, while one sample was collected via nasal swab, the second was collected from a mix of nasal swab and throat samples; this could conceivably introduce a greater variance into the allele frequencies.*

The reviewer is correct, and this is a challenging aspect of studies in natural infections with acute respiratory viruses. The standard deviation in variant frequency is indeed from a benchmarking experiment with in vitro material. However, given that viruses were diluted at set frequencies in media similar to the universal transport media used in clinical specimens and at genome equivalents similar to the range used in the study, we think that this provides a reasonable estimate of error. We struggle to come up with a better experiment that would provide some sort of estimate of precision given the lack of an in vivo “gold standard” for comparison. We acknowledge that there could certainly be spatial stratification as the reviewer suggests. However, as above, we note that Figure 2—figure supplement 1 shows that replicate same-day samples from 6 individuals separated by hours – one nasal and one nasal/throat – were well correlated with respect to iSNV type and frequency. For comparison, the excellent work of Lakdawala et al., cited by the reviewer reports spatial differences in allele frequencies for 3 ferrets who were experimentally infected.

[Editors' note: the author responses to the re-review follow.]

Essential revisions:This paper is a resubmission of a previously rejected version and was reviewed by the same reviewers. All reviewers agreed (as before) that the results on the transmission bottleneck are important, but two major concerns remain.1) Limited power to characterize within-host selection. Since most infections seem to be dominated by a single founder virus, the power to detect within-host selection during a few days of observation is very low. Selection would need to amplify a novel beneficial variant (produced at rates of about 1e-5 per day) to above 1% within 3-4 days in order to be detected in this study. Given the difficulty to call variants below 2%, even strong positive selection (s~10%) remains undetectable. Despite this lack of power, the authors make strong claims about the absence of positive selection and often make semi-quantitative statements without a proper comparison to an expectation.

We agree with this critique and recognizing that “absence of evidence is not evidence of absence,” we were careful in our manuscript to indicate that while positive selection is an inefficient driver over the course of an acute infection, it is definitely not absent. We concede that due to well-known issues in variant calling in NGS data, we only really make our arguments based on variants present at greater than 2% frequency and that this imposes a limit on what we can say about positive selection in an acute infection.

In response to the suggestions from these same reviewers from the previously rejected manuscript, we included additional analyses to look for evidence of selection with the suggested null models. First, we compared the number of HA variants in antigenic and non-antigenic sites (the former being more likely to be subject to positive selection) and found that our data were consistent with the null expectation of no difference (Figure 2B). Second, we asked whether any of the antigenic variants identified as minority SNV in our study were subsequently observed globally, as this would support the idea that they were “on the way up” due to positive selection. We found that this did not appear to be the case (Figure 2C). Finally, with the additional data presented now regarding the accuracy of our frequency measurements (see below), we can state more strongly that the data in Figure 2E regarding the dynamics of iSNV within hosts show evidence for considerable stochasticity. Here, the null expectation is that synonymous variants will show similar dynamics as antigenic variants (the latter being most likely to be affected by positive selection). We note that the number of variants detectable at >2% is roughly consistent with a neutral model based on influenza’s known mutation rate and generation time (see for example Box and figures in Xue et al., 2018.

In sum, we have performed several distinct analyses to identify evidence for positive selection in acute influenza infections, each with a reasonable null model. From these, we concluded that while positive selection clearly happens within hosts, it is not sufficiently strong to frequently drive new variants to fixation or even to levels where they are likely to be transmitted (our data would estimate that the probability of transmission is approximately 1.7% for a variant at 1% frequency and 3.3% for a variant at 2% frequency in both presence-absence and β-binomial models). Defining the limits of positive selection vis a vis drift at the level of individual hosts is an important aspect of our work, as three recent reviews on influenza evolution offer differing views on this question. (see Johnson et al., 2017; Petrova and Russell, 2017; Xue et al., 2018).

*a) Subsection “Mutations in antigenic epitopes are rare”. While you claim stochastic processes dominate within host and positive selection is inefficient, we don't think you have sufficient evidence for this statement. What you could conclude is "positive selection rarely amplifies a beneficial* de novo *variant above the limit of detection at 2%".*

We agree with this statement and this is largely how we discuss the data. We concede that our power to detect positive selection during acute selection is limited by the frequency of sampling and the reliability of variant detection below 2%. To address these concerns and limitations, we have further clarified our arguments and toned down any arguments regarding positive selection, see the following sentences (which include nearly all references to positive selection in the manuscript).

Introduction “Despite limited evidence for positive selection, novel mutations do arise within hosts and some will clearly be positively selected. Their potential for subsequent spread through host populations is heavily dependent on the size of the transmission bottleneck (Alizon et al., 2011; Zwart and Elena, 2015).”

Introduction “We find that intrahost populations are dynamic and constrained by genetic drift and purifying selection. In our study, positive selection rarely amplifies a beneficial de novo variant to a frequency greater than 2%.”

Subsection “Within host populations are dynamic” “Together with our prior work (Debbink et al., 2017), these data suggest that the positive selection of novel variants within hosts is inefficient and rarely amplifies a newly generated variant to a frequency greater than 2%.”

Subsection “Identification of forty-three transmission pairs” “Our within-host data suggest that newly arising iSNV with positive fitness effects are likely to be present at low frequencies (<2%) during an acute infection.”

Discussion section “Contrary to previous studies, which have found signatures of adaptive evolution in infected hosts (Gubareva et al., 2001; Rogers et al., 2015; Ghedin et al., 2011; Sobel Leonard et al., 2016), we have found only limited evidence of positive selection during acute infection. Previous reports have relied on infections in which selective pressures are likely to be particularly strong (e.g. due to drug treatment or infection with a poorly adapted virus), or in which the virus has replicated within a host for an extended period of time (Xue et al., 2017). Under these conditions, it is plausible that positively selected alleles may reach levels of detection. We suggest that these are important and informative exceptions to dynamics defined here, in which positive selection is rarely strong enough to drive a new mutation to a frequency above 2% over the course of several days.”

Discussion section “We have deeply sequenced 332 intrahost populations from 283 individuals collected over more than 11,000 person-seasons of observation and only identified a handful of minority antigenic variants with limited evidence for positive selection (this work and (Debbink et al., 2017)). Importantly, our data suggest that even if selection acts below our level of detection, such rare variants are unlikely to transmit. Given the size of the estimated bottleneck, the probability of transmission is approximately 1.7% for a variant at 1% frequency and 3.3% for a variant at 2% frequency. However, with several million infected individuals each year, even inefficient processes and rare events at the scale of individual hosts are likely to occur at a reasonable frequency on a global scale.”

b) In subsection “Within-host populations have low genetic diversity”, you are comparing R^2 values of variant frequencies of same day samples with the fraction of minor variants recovered in later samples. But an R^2 value and the fraction recovered iSNV are not comparable. To make a convincing case that your sequencing method accurately measures iSNV frequency and that the observed frequency changes are biological we would like to see the equivalent of Figure 2—figure supplement 1 for duplicate samples, ideally stratified by viral load (in subsection “Illumina library preparation and sequencing”, you write that samples were processed in duplicate). Such a control is particularly important for the comparison of iSNV frequencies in longitudinal samples, as the samples that are being compared likely differ in viral load. Figure 2—figure supplement 1 could use a 1:1 line. It seems that iSNV frequencies are systematically lower in-home isolates than in clinic isolates for low-frequency variants.

We appreciate these suggestions and have provided the requested analyses. We have added a 1:1 line to Figure 2—figure supplement 1A. There are several variants that were identified above the 2% cutoff in one sample but not the other. Together with the missing 1:1 line, this gave the appearance of a systematic bias in frequency measurement. As suggested, we have supplied a new Figure 2—figure supplement 1B, which shows iSNV frequency measurements across all duplicate samples stratified by viral load. Note that samples above 10^5^ copies per microliter were not processed in duplicate and those with fewer than 10^3^ copies per microliter were not used. This figure shows that our sequencing method accurately measures iSNV frequency. Finally, we have supplied a new Figure 2—figure supplement 1C, which compares the measurement error in frequency measurement (from duplicate RT-PCR and sequencing libraries) to the observed frequency differences in the serial biological samples. The impact of measurement error is minor. Together, these data strengthen our case regarding the biological significance of the changes in frequency in serial samples. They establish the bounds of what can reasonably be explained by stochastic and selective forces within hosts.

c) Given the difficulty to characterize within host stochasticity and the absence of positive selection, the claims made in this direction need to be revisited, toned down or removed. Furthermore, we suggest changing the title. A title like "Small transmission bottlenecks and low genetic diversity in acute influenza A virus infection". We are open to suggestions here, but please make sure that the title reflects what can be concluded from your data. On larger spatio-temporal scales, influenza virus evolution is certainly not stochastic and the current title doesn't make this distinction clearly enough.

As above, we have toned down where appropriate. We feel that our controls in variant calling and measurement error, our analyses of antigenic variants, and our quantitative analysis of SNV dynamics in serial samples demonstrate that while present, positive selection is an inefficient driver of variant accumulation within hosts – largely due to the short time scales (as summarized by the editor above). As we wrote in the Introduction of the submitted manuscript, “Despite limited evidence for positive selection, novel mutations do arise within hosts. Their potential for subsequent spread through host populations is heavily dependent on the size of the transmission bottleneck (Alizon et al., 2011; Zwart and Elena, 2015).” Therefore, we believe the within host data are important to the field, as they allow us to put the bottleneck in context, beyond simply providing a number. The data in Figure 2E suggest that variants arising through neutral processes are just as likely to be present at frequencies above 2% and therefore transmitted. Furthermore, the small size of the bottleneck and the good fit of a neutral model of transmission provide strong evidence for stochasticity. An alternative title that might more concisely capture the impact of stochasticity without overstating its role would be, “Stochastic processes *constrain* the within and between host evolution of influenza virus.” This title also clearly delineates the scale at which these processes are studied (individual hosts). We note that stochasticity in influenza evolution has been documented at larger scales (bigger than hosts, smaller than globally) in a paper by Nelson et al., “Stochastic processes are key determinants of short-term evolution in influenza”, 2006.

2) Confidence intervals on the transmission bottleneck size. You estimate the parameter of a Poisson distribution that is assumed to describe the distribution of bottlenecks in the population. The error bars of this estimate are remarkably tight given the small number of informative iSNVs in individual pairs. The actual distribution of bottleneck sizes in the population is certainly much broader and fitting a Poisson distribution will underestimate the confidence interval. Please include the 95% confidence intervals for the estimate of each transmission pair in Supplementary file 3. Aggregating these confidence intervals or fitting a broader distribution (say log-normal or power-law) should result in more sensible confidence intervals.

We have added the 95% confidence intervals (using the β binomial model) for the estimate of each transmission pair in Supplementary file 3. As expected, they are quite large because each transmission pair has so few data points. We have aggregated these confidence intervals using a log-normal distribution as suggested and weighted each transmission pair by the number of donor iSNV as performed in Sobel Leonard et al., 2017 (the same study in which the β binomial model was first implemented). The associated text now reads, “We also fit a long-tailed, discrete distribution based on the lognormal. As expected, this analysis resulted in a slightly wider distribution with a mode of 1, a 95^th^ percentile of 11 and a 97.5^th^ percentile of 21.” This analysis was run up to a maximum bottleneck of 1000. We have also added a curve to the bottleneck models in Figure 3 showing the probability that a variant will be transmitted given its frequency in the donor and a bottleneck of 10. This gives an indication of the fit of our aggregate bottleneck to the data.

In addressing this comment and those above, we noticed a “bug” in our code for plotting the β-binomial model in Figure 4E (now Figure 3E). The estimates and model fit as reported in the text were correct, but the plot was not. We have replaced this panel with the correct plot and apologize for our error. Consistent with the similarities in the results obtained through the presence-absence and β-binomial models, this plot is quite similar, but not identical to the one in Figure 3D.

3) Dependence of bottleneck estimates on sample pairings. We suggest, to estimate, bottlenecks from all possible different sample-pairings of donor and recipient in those transmission pairs where either has more than one sample available. In particular the difference between self-sampling and sampling in the clinics (Figure 2—figure supplement 1 suggests a possible systematic deviation) should be tested and discussed.

In the submitted manuscript we used the sample (home vs. clinic) that was obtained closer to the estimated time of transmission (based on onset of symptoms in the recipient). If the sample closest to the time of transmission had no minority SNV above 2%, we used the other sample (this is more conservative as we attempted to use all possible SNV data for these pairs). While we do not think that the data in Figure 2—figure supplement 1Ashow a systematic deviation, we do acknowledge that the choice of sample for these paired isolates from the 2014-2015 season entails certain assumptions about the transmission process. We have provided bottleneck estimates for all four possible combinations for which there are paired samples for each member of a pair as a source data file (Figure 3—source data 7).

We suggest resubmitting this manuscript as a short report focusing on transmission bottlenecks. Figure 4 is a plausible supplement to Figure 3.

We have moved Figure 4 to the supplement (now Figure 3—figure supplement 1) as suggested.